# CatScreen: A Large MultiModal Benchmark Dataset for Cataract Screening

**Mahapara Khurshid**                                                    *mahapara.1@iitj.ac.in*
*Indian Institute of Technology Jodhpur, India*

**Sonam Kumar**                                                    *sonamkumar.sk95@gmail.com*
*Postgraduate Institute of Medical Education and Research (PGIMER), Chandigarh, India*

**Anusuya Bhattacharyya**                          *anusuyabhattacharyya@aiimsguwahati.ac.in*
*All India Institute of Medical Sciences (AIIMS), Guwahati, India*

**Dhruve Kiyawat**                                                    *dhruvekiyawat739@gmail.com*
*Indian Institute of Technology Jodhpur, India*

**Anshul Chauhan**                                                    *anshsunny88@gmail.com*
*Postgraduate Institute of Medical Education and Research (PGIMER), Chandigarh, India*

**Suklengmung Buragohain**                                        *suklen.bgohain@gmail.com*
*Sri Sankaradeva Nethralaya (SSDN), Guwahati, India*

**Harsha Bhattacharjee**                                                    *ssnghy1@gmail.com*
*Sri Sankaradeva Nethralaya (SSDN), Guwahati, India*

**Limalemla Jamir**                                              *limajamir@aiimsguwahati.ac.in*
*All India Institute of Medical Sciences (AIIMS), Guwahati, India*

**Vishali Gupta**                                                    *vishalisara@yahoo.co.in*
*Postgraduate Institute of Medical Education and Research (PGIMER), Chandigarh, India*

**Mona Duggal**                                                    *monaduggal2@gmail.com*
*Postgraduate Institute of Medical Education and Research (PGIMER), Chandigarh, India*

**Mayank Vatsa**                                                    *mvatsa@iitj.ac.in*
*Indian Institute of Technology Jodhpur, India*

**Richa Singh**                                                    *richa@iitj.ac.in*
*Indian Institute of Technology Jodhpur, India*

**Reviewed on OpenReview:** *https://openreview.net/forum?id=cF7tSNAVQ6*

## Abstract

Low-cost slit-lamp imaging holds significant potential for transforming eye care by facil-itating affordable and scalable cataract diagnosis. However, the development of robust, generalizable AI-based cataract screening solutions is currently constrained by the limited availability of large-scale, richly annotated datasets. To address this critical gap, we intro-duce CatScreen, a comprehensive multimodal benchmark dataset specifically designed for cataract screening, comprising approximately 18,000 slit-lamp images collected from 2,251 subjects using a portable slit-lamp camera. CatScreen is structured into three subsets: (i) a clean set meticulously annotated using a structured multi-tier framework involving trained optometrists with final validation by an experienced ophthalmologist across clinically rele-

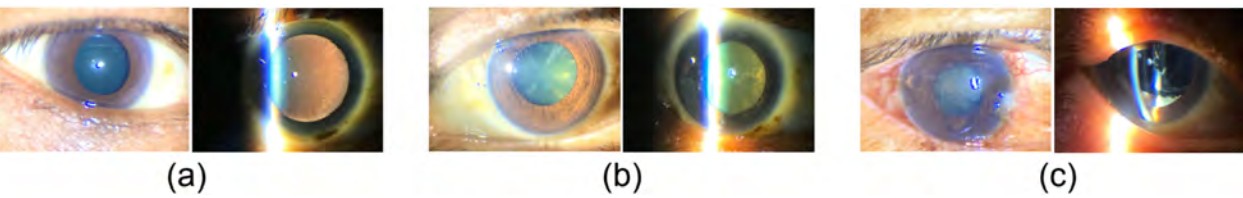

Figure 1: Sample images of eyes for (a) normal, (b) cataract condition characterized by lens clouding, and (c) various other eye conditions affecting different ocular structures.

vant dimensions, including image gradability, quality assessment, illumination type, diagnostic classification, cataract subtype, and severity grading according to established standards; (ii) a noisy-labeled set that simulates real-world annotation inaccuracies; and (iii) an unlabeled set intended to foster the development of self-supervised and semi-supervised learning approaches. Furthermore, CatScreen integrates extensive subject-level metadata encompassing demographics, lifestyle factors, and detailed clinical histories, and includes a subset with anatomical and pathological annotations to support multimodal modeling and anatomically grounded analysis. We present baseline experiments under independent, structured sequential, and multitask prediction settings in both unimodal and multimodal configurations. These results establish initial benchmarks for CatScreen and demonstrate the value of metadata for selected diagnostic tasks, while also highlighting open challenges, such as class imbalance and fine-grained subtype discrimination. CatScreen is intended as a benchmark resource for future research in cataract screening, robust learning, semi-supervised learning, and interpretability-oriented analysis. The database is available at: https://iab-rubric.org/resources/healthcare-datasets/catscreen.

# 1 Introduction

Cataract is characterized by clouding of the eye's natural lens, resulting in diminished visual acuity. If left untreated, cataracts can progress to complete blindness or significantly impair an individual's mental well-being Wang et al. (2024b). Figure 1 illustrates representative examples of cataractous lens compared to healthy eyes and natural crystalline lens affected by other ocular conditions. Globally, cataract remains the predominant cause of blindness, impacting approximately 15.2 million individuals, constituting about 45% of global blindness cases (Steinmetz et al., 2021; of the Global Burden of Disease Study et al., 2024). Although cataracts have traditionally been prevalent among elderly populations, recent epidemiological studies report a worrying rise in prevalence among younger age groups. This demographic shift is associated with multiple modifiable and non-modifiable risk factors, including smoking, radiation exposure, and an increasing incidence of diabetes (Thompson & Lakhani, 2015). Timely detection and early surgical intervention through lens replacement remain the definitive treatment options, thereby emphasizing the necessity of effective and accessible screening methods.

Conventionally, cataract diagnosis relies heavily on manual examinations performed by trained ophthalmologists using slit-lamp microscopy. This clinical evaluation requires significant expertise and extensive practical experience to accurately detect cataracts, classify their subtypes, and grade their severity. Despite its effectiveness, this traditional diagnostic approach faces substantial limitations, particularly in resource-limited settings characterized by insufficient qualified healthcare personnel (Flaxman et al., 2017). Consequently, a significant fraction of the global population, especially those in underdeveloped and underserved regions, lacks access to timely diagnosis and effective treatment options. This disparity underscores the urgent need for innovative, scalable, and accessible diagnostic solutions that can bridge existing healthcare gaps and enhance ocular health outcomes for diverse populations.

Recent advancements in Artificial Intelligence (AI) and Machine Learning (ML) present promising avenues to address these diagnostic challenges. AI techniques have shown substantial potential in medical imaging analysis, particularly in detecting, segmenting, classifying, and grading various ocular diseases (Gour et al., 2023).

Table 1: Existing slit-lamp image-based datasets for cataract diagnosis and their respective characteristics

| Author | Device Used | No. of Subjects | No. of Images | Metadata | Image Quality? | Disease Grading? | Remarks |
|---|---|---|---|---|---|---|---|
| Foong et al. (2007) | Topcon DC-1 digital slit-lamp | 1000 | 1000 | ✓ | ✗ | ✓ | Nuclear Cataract |
| Jiang et al. (2021) | slit-lamp | 536 | 886 | ✗ | ✗ | ✓ | Infantile Cataract Single Center |
| Jiang et al. (2021) | slit-lamp | 433 | 757 | ✗ | ✗ | ✓ | Infantile Cataract Multicenter |
| Son et al. (2022) | SL-D7[TMS, Inc] D850 [Nikon, Inc] | 596 | 1972 | ✓ | ✗ | ✓ | Cataract detection and grading |
| Wang et al. (2024a) | slit-lamp | 2150 | 2150 | ✗ | ✗ | ✓ | Cortical cataract |
| Shimizu et al. (2023) | Smart Eye Camera | 1812 | 38320 frames | ✗ | ✗ | ✗ | Nuclear cataract |
| **Proposed CatScreen** | Remidio Portable Slit Lamp PSL-D20 | 2251 | 18640 | ✓ | ✓ | ✓ | Cataract detection, grading, quality analysis, multimodal data |

Integrating AI-driven automated systems into cataract screening workflows can significantly enhance diagnostic accuracy, efficiency, and accessibility, offering profound benefits to remote and resource-constrained environments. However, the development and successful deployment of robust AI-based diagnostic tools fundamentally depend on the availability and quality of comprehensive, diverse, and meticulously annotated datasets.

Several research initiatives have leveraged slit-lamp imaging datasets to automate cataract detection and classification. However, current slit-lamp datasets often suffer from limitations related to data diversity, completeness, and annotation depth, hindering their applicability to generalizable and robust diagnostic systems. For instance, Foong et al. (2007) and Shimizu et al. (2023) focused primarily on nuclear cataracts, while Wang et al. (2024a) concentrated exclusively on cortical cataracts. Such narrowly focused datasets inherently lack comprehensive diversity, thus limiting their generalizability. Similarly, datasets provided by Jiang et al. (2021) and Son et al. (2022), although extensive, primarily offer limited annotation dimensions, restricting the clinical relevance and interpretability of AI solutions derived from them. Notably absent from these datasets are critical parameters such as detailed assessments of image quality, precise classification of cataract subtypes, and standardized grading of cataract severity, all of which are essential elements in comprehensive ophthalmic diagnosis. Consequently, existing AI-based methods developed from such datasets encounter significant biases and constraints in real-world diagnostic scenarios.

To systematically address these limitations and foster the development of robust and clinically relevant AI-based cataract screening tools, we introduce CatScreen, an extensive, multimodal, and comprehensively annotated dataset acquired through an affordable, portable slit-lamp imaging device. While conventional tabletop slit lamps offer superior optical stability, their cost, size and infrastructure requirements limit deployment in outreach and resource-constrained settings (Kaushik et al., 2026). In contrast, smartphone-integrated portable slit-lamp systems enable scalable and accessible cataract screening (Liu et al., 2026). However, these systems introduce increased variability in image quality due to factors such as illumination inconsistency, motion artefacts, and operator dependence (Fung et al., 2024). CatScreen is designed to reflect community-level screening conditions, including variability in image quality, illumination, and operator dependence, thereby providing a realistic benchmark for studying robustness and generalization. The dataset comprises over 18,000 slit-lamp images collected from approximately 2,251 diverse subjects. Each image is thoroughly annotated across several clinically relevant dimensions, including diagnostic classification, image gradability, quality assessments, cataract subtype identification, and severity grading, all adhering to established ophthalmological standards. Additionally, comprehensive subject-level metadata capturing essential demographic variables, lifestyle factors, and clinical histories accompanies the dataset. In addition, a subset of images includes detailed region annotations such as the lens, cornea, iris, and pathological regions that support anatomically grounded analyses and future interpretability-oriented modelling.

By providing a richly annotated and multidimensional ocular dataset, CatScreen uniquely addresses critical shortcomings of existing datasets and sets a foundational resource for benchmarking multimodal cataract screening models and supporting future work in explainability, robust learning, and clinically motivated analysis. The depth and diversity of annotations ensure that AI methods developed using CatScreen can support development and benchmarking across diverse acquisition settings, enhancing their utility in real-world ophthalmic diagnostics. Given these comprehensive features, CatScreen directly addresses the following critical clinical and technical research questions:

**Q1**: How can slit-lamp image quality be reliably and objectively evaluated?

**Q2**: How can the type of illumination employed in slit-lamp imaging be accurately identified, and its impact on diagnostic outcomes quantified?

**Q3**: How can slit-lamp images be effectively categorized into normal, cataractous, or other ocular conditions?

**Q4**: How can specific cataract subtypes be accurately identified and differentiated?

**Q5**: Upon diagnosing cataract presence, how can cataract severity be consistently and objectively assessed?

The structured and comprehensive nature of CatScreen ideally positions it for diverse screening and diagnostic modeling frameworks, including structured sequential analyses and multitask learning methodologies, as elaborated in the subsequent problem formulation section.

## 2 Problem Formulation

The problem addressed in this work is the automated screening and diagnostic assessment of cataracts, aiming to replicate the detailed multi-stage clinical evaluation typically performed by ophthalmologists. Specifically, this assessment involves evaluating the quality of slit-lamp images, classifying illumination conditions, diagnosing the presence or absence of cataracts, identifying specific cataract subtypes, and grading cataract severity. The structured assessment utilizes slit-lamp images paired with comprehensive patient metadata to provide clinically informed predictions.

Formally, we denote an input slit-lamp image as $I \in \mathbb{R}^{H \times W \times C}$, where $H$, $W$, and $C$ correspond to image height, width, and color channels, respectively. Additionally, patient metadata is represented by a feature vector $M \in \mathbb{R}^d$, containing demographic details, lifestyle factors, and clinical histories, such as age, gender, diabetic status, and ocular medical records. The task objective is to predict a set of clinically relevant diagnostic attributes, denoted as $A = (Q, T, D, C, G)$, where each attribute represents:

- $Q \in$ Good, Acceptable, Poor: Image quality assessment.

- $T \in$ Diffuse, Direct Focal, Retro: Illumination type classification.

- $D \in$ Normal, Cataract, Others: Diagnostic categorization.

- $C \in$ No Cataract, Nuclear, Cortical, PSC, Pseudophakia, Others: Cataract subtype identification.

- $G \in$ Not Applicable, Mild, Severe: Severity grading of cataract.

This diagnostic formulation can be approached via two distinct yet complementary methodologies: (1) structured sequential analysis, mirroring the sequential decision-making workflow of clinical practitioners, or (2) multitask learning, where all diagnostic attributes are concurrently predicted from shared inputs.

To reflect the clinically accepted ordering of decisions in cataract screening, we formulate a structured sequential dependency model, Structured Sequential Analysis (SSA), over the target attributes. The model is formally defined as follows:

$$Q = f_Q(I, \epsilon_Q), \tag{1}$$
$$T = f_T(I, Q, \epsilon_T), \tag{2}$$
$$D = f_D(I, M, Q, T, \epsilon_D), \tag{3}$$
$$C = f_C(I, M, D, \epsilon_C), \tag{4}$$
$$G = f_G(I, M, D, C, \epsilon_G), \tag{5}$$

where $\epsilon_Q, \epsilon_T, \epsilon_D, \epsilon_C, \epsilon_G$ denote exogenous noise variables, thus providing a structured representation of task dependencies that supports more transparent discussion of the diagnostic workflow.

Alternatively, the multitask learning formulation employs a unified predictive model $f_\theta$ parameterized by model weights $\theta$, simultaneously predicting all attributes:

$$(\hat{Q}, \hat{T}, \hat{D}, \hat{C}, \hat{G}) = f_\theta(I, M). \tag{6}$$

This model is optimized by minimizing a weighted sum of task-specific classification losses:

$$\mathcal{L}(\theta) = \lambda_Q \mathcal{L}_Q + \lambda_T \mathcal{L}_T + \lambda_D \mathcal{L}_D + \lambda_C \mathcal{L}_C + \lambda_G \mathcal{L}_G, \tag{7}$$

with each individual loss $\mathcal{L}$. defined by categorical cross-entropy:

$$\mathcal{L}(y, \hat{y}) = -\sum_{j=1}^{K} y_j \log(\hat{y}_j), \tag{8}$$

where $y$ is the one-hot encoded ground-truth vector, and $\hat{y}$ is the predicted probability distribution over $K$ classes.

Acknowledging the advantages inherent in each modeling approach, we explicitly structured our CatScreen dataset to facilitate both methodologies. This flexibility ensures the development of robust, versatile, and clinically meaningful automated cataract screening systems.This sequential formulation draws on established clinical workflows to provide a structured prediction framework suited for benchmarking, without imposing formal causal assumptions on the relationships among diagnostic attributes.

## 3 CatScreen Dataset

The CatScreen dataset is a meticulously curated resource, comprising over 18,000 slit-lamp images acquired using low-cost, portable handheld slit-lamp cameras. Aiming to deliver a comprehensive and affordable solution for cataract diagnosis, this dataset addresses critical gaps identified in existing ophthalmic image repositories. Prioritizing patient privacy and adhering strictly to ethical guidelines, all Protected Health Information (PHI), including names, guardian details, contact numbers, email addresses, and other sensitive data, has been carefully anonymized and removed. To maintain data integrity without compromising confidentiality, each image is assigned a unique, non-identifying identifier indicating patient ID, pupil dilation state (dilated or undilated), eye side (right or left), and capturing device details. Preliminary preparations, including rigorous site selection, commenced in August 2022, leading to an extensive data collection phase spanning from December 2022 through January 2024. This section elaborates on the detailed prerequisites, device specifications, data acquisition procedures, and the exhaustive annotation methodologies utilized.

### 3.1 Pre-requisites and Device Specifications

This subsection describes the essential preparatory steps, patient recruitment protocols, exclusion criteria, and technical specifications of the imaging device employed.

1. **Ethical Considerations**: Prior to data collection, formal approvals were obtained from the Institutional Ethics Committees of the Postgraduate Institute of Medical Education and Research (PGIMER) (approval number PGI/IEC-11/2022-2633) and Sri Sankaradeva Nethralaya (SSN/IEC/AUGUST/2022/01). Adherence to the ethical standards outlined in the Declaration of Helsinki was strictly maintained throughout the study. Participants were given detailed briefings on the study's objectives, methodologies, and potential implications. Written informed consent was obtained from all participants, and dedicated sessions were held to address and resolve the questions of the participants.

2. **Exclusion Criteria**: To guarantee data consistency, individuals with active ocular infections or diagnosed type 1 diabetes mellitus were excluded from participation. These criteria ensured the collection of relevant and high-quality data suitable for developing reliable diagnostic models.

3. **Device Specifications**: The imaging data was captured using the Remidio Portable Slit Lamp PSL-D20 (Remidio Innovative Solutions Pvt. Ltd., Bangalore, India), a handheld, versatile, and portable digital slit-lamp camera. Designed explicitly for efficient mobile eye examinations, the PSL-D20 device offers adjustable slit widths ranging from 1 to 12mm and slit angles adjustable within $\pm45$ degrees, supporting both diffuse and direct focal illumination. The device seamlessly integrates with an Apple iPhone SE camera, leveraging its high-resolution imaging capabilities. It supports multiple specialized imaging modes, including standard color imaging, green (red-free), and black illumination. Equipped with magnification options of $10\times$ and $16\times$, and a working distance of 61mm, the PSL-D20 ensures high-quality imaging under diverse clinical settings. Images are captured at a high resolution of $3024\times4032$ pixels, while videos are recorded at $1920\times1080$ pixels, further enhancing data clarity and clinical applicability.

### 3.2 Data Collection

The CatScreen dataset comprises 18,640 slit-lamp images from 2,251 diverse participants collected across multiple clinical sites. In addition to visual imagery, CatScreen includes comprehensive metadata capturing detailed socio-demographic and clinical characteristics of each participant, significantly enriching the dataset's analytical potential. To facilitate various research approaches, CatScreen is systematically categorized into three distinct subsets:

- **Clean Set**: Featuring 9,915 meticulously annotated images from 1,271 participants, this subset serves as the foundation for robust, supervised learning model training. Domain experts provided exhaustive annotations covering essential parameters including image gradability, diagnostic categorization, cataract subtype identification, severity grading, and illumination type. The detailed distribution of annotations across patients and attributes is presented in Table 2a.

- **Noisy Set**: Consisting of 3,267 images from 505 participants, this subset includes annotations with varying degrees of accuracy to emulate realistic clinical environments, where annotations may inherently contain errors. This set is annotated solely at Level 1 without subsequent review by annotators at Levels 2 and 3. Since level 1 annotations may contain errors due to differences in expertise and diagnostic precision, this subset inherently incorporates real-world label noise, particularly representative of community-based screening contexts where specialist review is often unavailable. Such a dataset aids in the development and evaluation of models resilient to annotation noise.

- **Unlabelled Set**: Containing 5,481 images without diagnostic annotations, this subset facilitates the exploration of innovative approaches such as self-supervised, semi-supervised, and active learning methodologies, allowing effective utilization of unlabeled data to augment model robustness and generalizability.

### 3.2.1 Image Collection Protocol

Images were systematically captured following the protocols recommended by the WHO simplified grading system. These images were collected under both dilated and undilated pupil conditions, using the following specialized imaging techniques:

- **Direct Optical Section Images**: Captured at $10\times$ magnification, with a slit width of 0.2 mm and a slit-beam orientation of 30 degrees, enabling detailed visualization of distinct lens layers.

- **Direct Diffuse Illumination Images**: Obtained at $5\times$ magnification, with the illuminating arm positioned at a 45-degree angle relative to the microscope. This approach provides comprehensive visualizations of the anterior segment of the eye in both dilated and undilated states.

- **Indirect Retro-Illumination Images**: Acquired under strictly controlled, dimly lit conditions (below 15 lux) with a slit lamp-mounted iPhone SE camera flash, exclusively in dilated conditions, offering enhanced visualization of posterior lens structures and subtle abnormalities.

### 3.2.2 Comprehensive Metadata

To maximize the clinical and research value of the CatScreen dataset, an extensive array of patient metadata was collected, encompassing a broad spectrum of factors potentially influencing cataract development and progression. Summarized comprehensively in Table 2b, key metadata elements include:

- **Demographic Data**: Participant ages ranged from 35 to 92 years, averaging approximately 57 years, with a well-balanced gender distribution comprising 45% females and 54% males.

- **Health Conditions**: Clinical histories documenting the presence of diabetes, hypertension, and other systemic conditions that could influence ocular health.

- **Eye Health History**: Detailed ocular health records capturing previous incidents of eye inflammation, myopia, ocular surgeries, eye trauma, and topical ocular disorders.

Through its careful curation, rigorous annotations, and extensive metadata, CatScreen provides a valuable benchmark resource for advancing AI-driven cataract screening research and supporting future clinically relevant studies, particularly for underserved and resource-limited settings.

## 4 Data Annotation

Ensuring accuracy and consistency in dataset annotation is pivotal for developing robust and clinically reliable diagnostic models. To achieve this objective, the CatScreen dataset employed a meticulous three-tiered annotation process, structured to maximize precision, minimize annotation biases, and maintain high-quality standards throughout. Image acquisition and initial annotation were conducted by formally trained optometrists who underwent structured institutional training in slit-lamp operation, ocular imaging, and quality assurance. Imaging consistency was maintained through standardized acquisition protocols, on-site supervision, and real-time quality checks prior to annotation.

### 4.1 Three-tiered Annotation Structure

The annotation procedure was designed to leverage progressively specialized expertise at each annotation level:

1. **Level 1 – Initial Annotation**: Two experienced optometrists, each holding a bachelor's degree in optometry with a minimum of three years of hands-on experience in retinal and anterior segment imaging, performed initial annotations. These annotations provided a foundational layer for subsequent refinement.

| Label | Classes | Train | Val | Test | Total Images |
|---|---|---|---|---|---|
| **Diagnosis** | Normal | 1845 | 251 | 545 | **2641** |
| | Cataract | 3627 | 523 | 996 | **5146** |
| | Others | 1526 | 224 | 378 | **2128** |
| **Cataract Type** | Not Applicable | 1845 | 251 | 545 | **2641** |
| | Nuclear Cataract | 3357 | 479 | 924 | **4760** |
| | Cortical Cataract | 184 | 30 | 48 | **262** |
| | Posterior SC | 75 | 14 | 24 | **113** |
| | Pseudophakia | 1495 | 220 | 365 | **2080** |
| | Others | 42 | 4 | 13 | **59** |
| **Cataract Grade** | Not Applicable | 3371 | 475 | 923 | **4769** |
| | Mild | 2970 | 428 | 796 | **4194** |
| | Severe | 657 | 95 | 200 | **952** |
| **Image Quality** | Good | 134 | 22 | 20 | **176** |
| | Acceptable | 6112 | 871 | 1680 | **8663** |
| | Poor | 752 | 105 | 219 | **1076** |
| **Illumination type** | Diffuse Illumination | 2130 | 296 | 606 | **3032** |
| | Direct Focal | 2996 | 423 | 795 | **4214** |
| | Retro | 1872 | 279 | 518 | **2669** |

(a)

| Set | | | Gender | | | Age | | |
|---|---|---|---|---|---|---|---|---|
| | Participants | | | Clean | Noisy | | Clean | Noisy |
| Clean Set | 1271 | | Females | 578 | 231 | Max | 92 | 86 |
| Noisy Set | 505 | | Males | 693 | 274 | Min | 35 | 51 |
| | | | | | | Mean $\pm$ Std | $57.2 \pm 9.96$ | $61 \pm 7.22$ |

| Diabetes Status | | | Hypertension Status | | | Systemic Illness Status | | |
|---|---|---|---|---|---|---|---|---|
| | Clean | Noisy | | Clean | Noisy | | Clean | Noisy |
| No | 815 | 199 | No | 710 | 181 | No | 1002 | 295 |
| Yes | 456 | 306 | Yes | 561 | 324 | Yes | 269 | 210 |

| Eye Problem Status | | | Use of Spectacles | | | Ocular Surgery Done? | | |
|---|---|---|---|---|---|---|---|---|
| | Clean | Noisy | | Clean | Noisy | | Clean | Noisy |
| No | 396 | 195 | No | 367 | 166 | No | 667 | 354 |
| Yes | 875 | 310 | Yes | 904 | 339 | Yes | 604 | 151 |

(b)

Table 2: Showcasing the (a) distribution of each label in training, validation and testing, and (b) summarizing the distribution of participants of clean and noisy sets based on sociodemographic factors and other clinical characteristics.

2. **Level 2 − Secondary Review and Refinement**: A senior optometrist possessing a master's degree in optometry and over five years of clinical and imaging experience conducted an exhaustive secondary review. This intermediate stage involved meticulous refinement and correction of initial annotations to resolve ambiguities and discrepancies.

3. **Level 3 − Expert Validation**: Final validation and oversight were provided by a retina-trained ophthalmologist with over a decade of clinical and diagnostic experience in slit-lamp–based anterior segment evaluation. This expert-level validation ensured maximum accuracy, consistency, and clinical relevance of the final annotations, thereby enhancing the dataset's reliability and applicability in developing diagnostic AI solutions.

The annotation process was systematically divided into three essential phases: gradability assessment, ground-truth collection, and anatomical and pathological annotation.

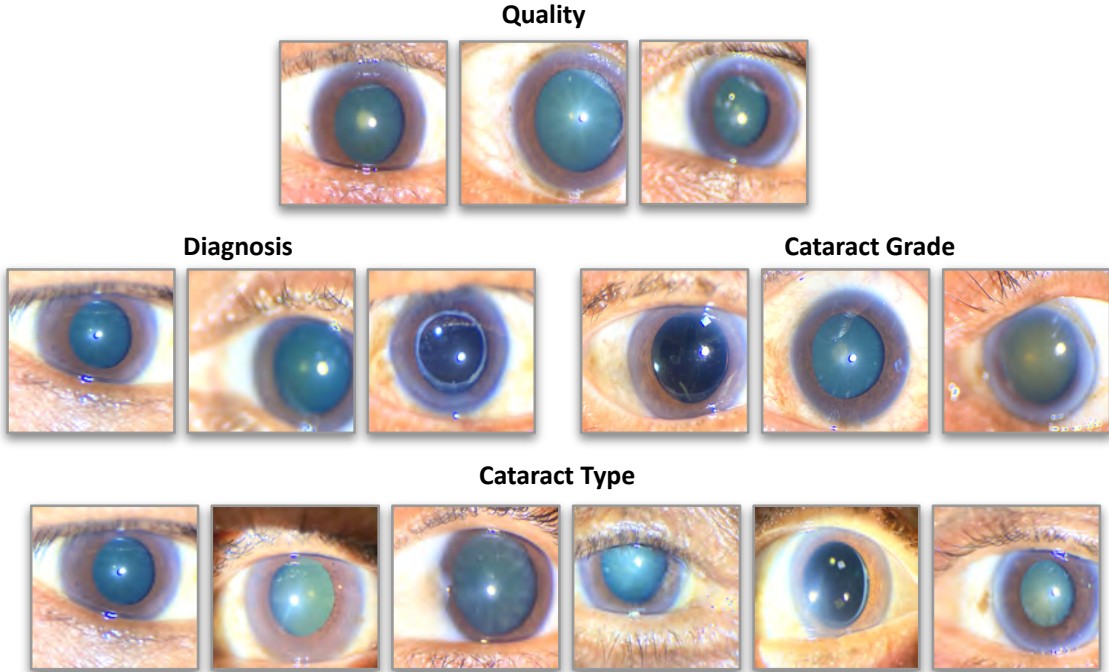

Figure 2: Presents the samples of the CatScreen in diffuse illumination setting across various labels.

## 4.2 Gradability Assessment

Image gradability is crucial for ensuring diagnostic accuracy in ophthalmic imaging research. Recognizing its importance, a rigorous two-stage gradability assessment protocol was developed and implemented.

In the first stage, trained opticians assessed the image quality in real-time at the point of capture. Images failing predefined stringent criteria related to clarity, sharpness, and contrast were immediately discarded and replaced with newly captured images. The second stage involved Level 1 annotators performing a comprehensive reevaluation of the images using identical quality standards. Only those images passing both stages of rigorous scrutiny were deemed gradable and included for detailed annotation and further analysis.

The key parameters evaluated during this gradability assessment included:

- Clear visibility of at least 80% of the anterior segment of the eye.
- Distinct visibility of critical morphological features including the iris, lens, cornea, and other anterior ocular structures.
- Optimal sharpness and contrast for accurate clinical interpretation.
- At least 95% clarity of all morphological landmarks.

## 4.3 Ground-truth Collection

Each selected image underwent a detailed multi-label annotation process, capturing comprehensive diagnostic insights. Figure 2 showcases the multi-label annotation of some sample images in diffuse illumination.

**a. Image Quality**: Annotated using a 10-point Likert scale, the image quality ratings were aggregated into three distinct categories to mitigate subjective variability: Poor (Grades 1–3), Acceptable (Grades 4–8), and Good (Grades 9–10).

**b. Illumination Type**: Images were categorized based on three illumination techniques commonly used in slit-lamp imaging:

- **Direct Focal Illumination**: Utilizing a narrow, focused beam directed to highlight specific anterior ocular structures such as the lens and cornea, enhancing visibility of fine details and layer structures.

- **Diffuse Illumination**: Employing a broad, uniform light source, this method provides an overall visualization of the anterior segment, suitable for general screening and holistic assessment.

- **Retro-Illumination**: Involving light reflected from behind the retina to visualize the posterior lens, this method accentuates subtle abnormalities such as posterior lens opacities.

**c. Diagnostic Label**: Images were classified into three diagnostic categories:

- **Normal**: Images with no detectable abnormalities.

- **Cataract**: Images exhibiting any evidence of lens clouding.

- **Others**: Images indicating conditions other than cataract, such as pseudophakia, pterygium, conjunctivitis, or corneal scarring. This grouping aggregates all non-cataract cases to align with the primary objective of cataract detection and severity assessment.

**d. Cataract Type**: Six categories were utilized for identifying cataract subtypes:

- **Not Applicable**: For cases with no visible opacification.

- **Nuclear Cataract**: Opacities predominantly within the lens nucleus.

- **Cortical Cataract**: Peripheral lens opacities characterized by wedge-like patterns.

- **Posterior Subcapsular Cataract (PSC)**: Opacities located posteriorly, affecting the lens capsule's back.

- **Pseudophakia (PP)**: Presence of an artificial intraocular lens.

- **Others**: Any conditions not fitting exclusively into above categories or combinations thereof.

**e. Cataract Grade**: Cataract severity was classified into three distinct grades:

- **Not Applicable**: For normal or other non-cataract cases.

- **Mild**: Early-stage cataracts with minimal to moderate lens opacity. This corresponds to clinical grades 1, 2, and 3, which are grouped to mitigate inter-observer variability and subjective bias inherent in fine-grained clinical grading.

- **Severe**: Advanced cataracts characterized by dense and extensive lens opacification. This corresponds to clinical grades 4 and 5.

### 4.4 Anatomical and Pathological Annotations

To further enhance the dataset's interpretability and clinical relevance, a subset of images underwent meticulous segmentation-based annotations. Using the Oxford VGG Image Annotator tool (available at [https://www.robots.ox.ac.uk/~vgg/software/via/via_demo.html]), annotators delineated key ocular structures such as the cornea, iris, lens, and sclera. Additionally, pathological features such as cataracts, corneal scars, pterygium, subconjunctival hemorrhages, and edema were accurately identified and annotated. These detailed annotations are critical for training AI models capable of anatomically accurate diagnostic predictions and for validating the clinical interpretability of the models. Sample annotated images are presented in Figure 3, and precise coordinates of each annotation are systematically documented.

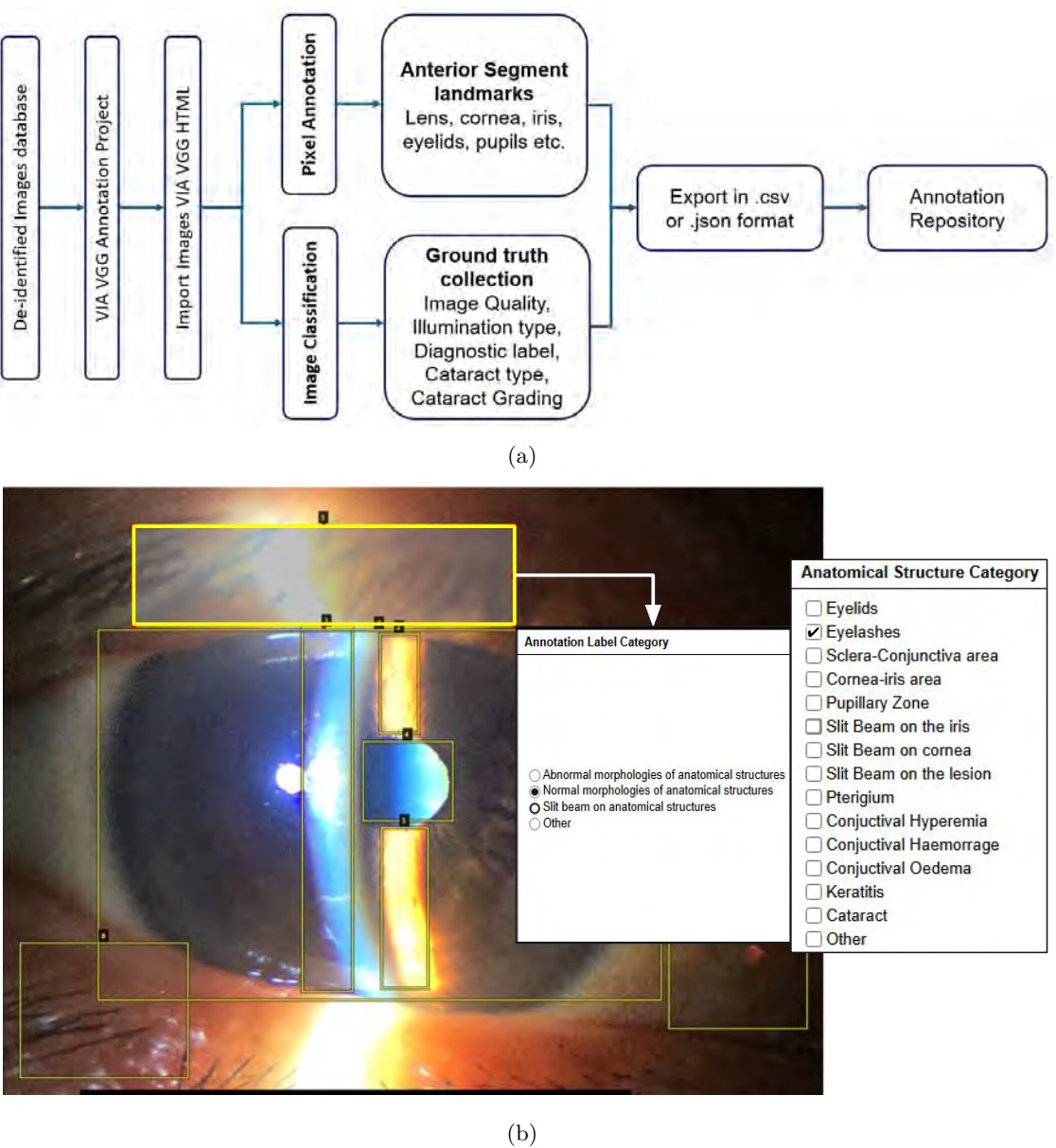

Figure 3: Presents (a) the annotation process where the input image is classified corresponding to multiple labels and (b) various annotations for anatomical and pathological regions in a slit-lamp image.

### 4.5 Quality Assurance Protocol

A comprehensive quality assurance (QA) protocol was implemented, involving initial annotations by Level 1 annotators followed by systematic reviews by Level 2 annotators. For initial QA validation, the first 500 annotated images underwent inter-annotator agreement analysis, specifically targeting diagnostic labels. A kappa score threshold of 0.60 was set as the standard for acceptable agreement. Any discrepancies or disagreements were rigorously addressed in structured group discussions involving the expert Level 3 annotator. Following this initial validation phase, continuous monthly QA checks were conducted, reviewing 25% of annotated images over three consecutive months. This rigorous annotation and QA protocol ensured high levels of annotation precision and consistency, reinforcing CatScreen's suitability as a reliable and clinically valuable dataset for developing advanced ophthalmic diagnostic AI solutions.

### 4.6  Data Records

The CatScreen dataset is available for research purposes and a sample snapshot of the dataset is available at https://iab-rubric.org/resources/healthcare-datasets/catscreen. There is a primary folder named CatScreen with the following directory structure:

- Clean: This folder includes a subfolder "Clean images" containing clean image data and a metadata file, *clean_meta*, with detailed socio-demographic and medical information for each participant in the clean set. Additionally, three CSV files for training, testing, and validation are provided, containing label information such as image quality, diagnostic label, severity grade, and more.

- Noisy: This folder includes a subfolder "Noisy images" containing noisy image data and a metadata file, *noisy_meta*, with detailed socio-demographic and medical information for each participant. A CSV file, *noisy_labels*, is also included, providing label information for each attribute.

- Unlabelled dataset: This folder includes a subfolder "Unlabelled images".

- A CSV file, "Region_annotation", is also included, containing the filenames and coordinate information of various regions within each image.

## 5  CatScreen Framework and Experimental Details

This section presents a detailed overview of the CatScreen Framework, aligning it with the mathematical problem formulation described in Section 2. It introduces two complementary modeling approaches, Structured Sequential Analysis and Multitask Learning, and provides descriptions of the evaluation protocols, baseline algorithms, and implementation specifics essential for validation of automated cataract screening solutions.

### 5.1  CatScreen Framework

The CatScreen Framework is designed to address the automated diagnostic tasks delineated mathematically in Section 2, specifically targeting the accurate prediction of clinically relevant attributes $A = (Q, T, D, C, G)$ from slit-lamp images $I$ and patient metadata $M$. To establish representative benchmark settings for CatScreen, we consider two paradigms: Structured Sequential Analysis and Multitask Learning.

1. Structured Sequential Analysis: Figure 4a represents the Structured Sequential Analysis framework. This approach closely mimics the clinical reasoning of the doctors by explicitly modelling the dependencies among diagnostic attributes. Each stage builds upon preceding predictions, reflecting the typical sequential decision-making of ophthalmologists.

   Starting with a slit-lamp image (I), the model first assesses the image quality (Q) and illumination type (T). These are important as they directly affect the subsequent diagnostic predictions. Based on these intermediate assessments, the model then predicts the overall diagnosis (D) by combining it with patient metadata (M), determining whether the eye is normal, has a cataract or shows other conditions. If a cataract is detected, the model then proceeds to classify the cataract type and its severity grade. By sequentially building each prediction upon prior outputs, this approach facilitates explicit feature refinement and error control at each stage, aligning well with medical interpretability requirements.

2. Multitask Learning Framework: Figure 4b presents the multitask approach. Unlike the sequential approach, this framework predicts all the diagnostic attributes jointly from the same set of inputs. A shared feature encoder first extracts global representations from the slit-lamp image (I), enriched by patient metadata (M), thereby capturing visual and contextual information. Separate task-specific classification heads are then used to simultaneously predict image quality (Q), illumination type (T), diagnosis (D), cataract subtype (C), and severity grade (G). This parallel formulation allows the model to leverage correlations among tasks while avoiding explicit sequential dependencies, thus

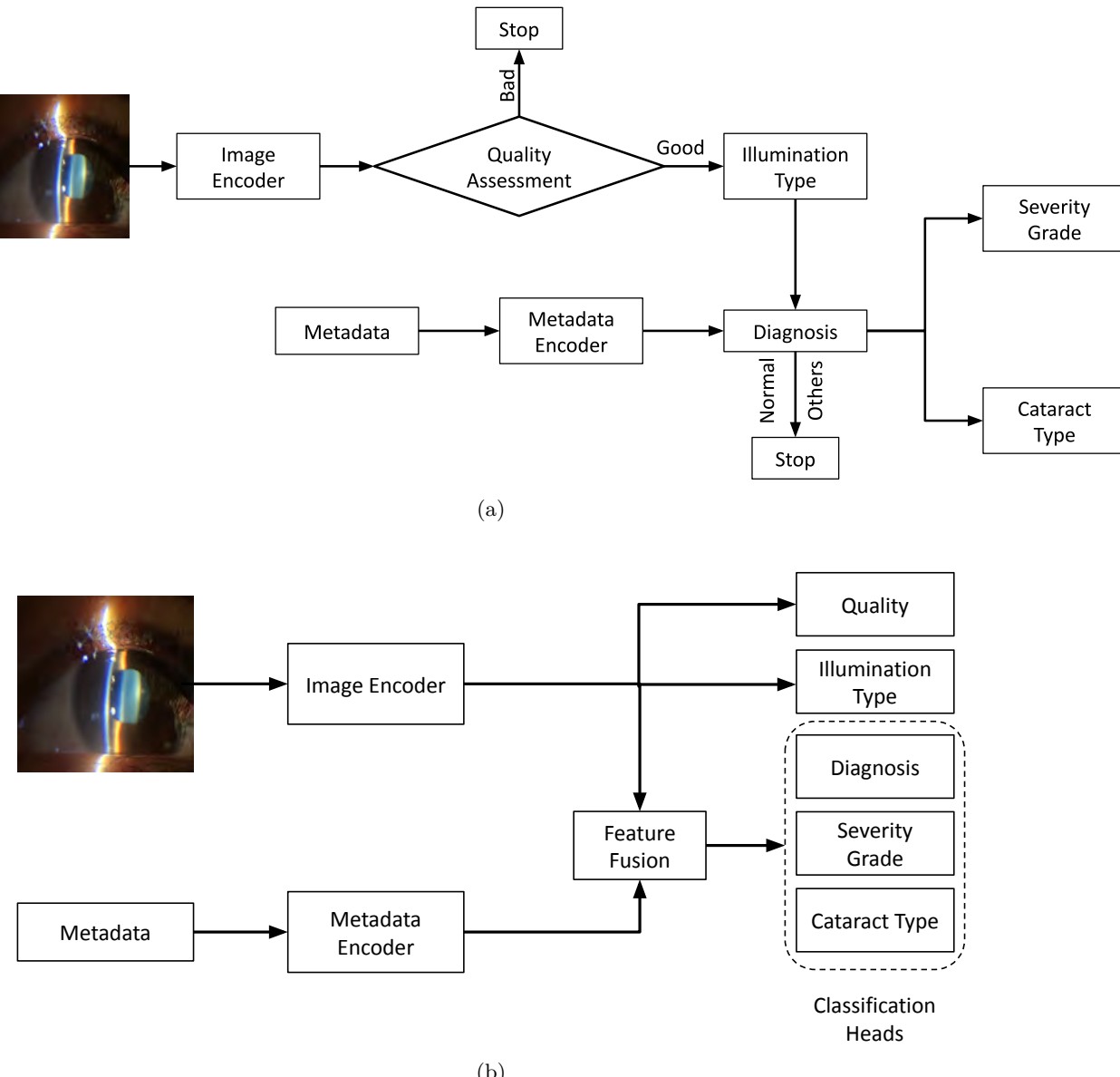

Figure 4: Presents the two complementary frameworks: (a) structured sequential analysis framework for automated cataract screening and (b) Multitask learning framework where each label is jointly predicted for automated cataract screening.

enabling efficient end-to-end learning. The overall training objective combines individual classification losses for each task into a composite loss function (as described in Equation 7), promoting shared representation learning while allowing specialization for each attribute.

## 5.2 Experimental Protocol and Baselines

To evaluate the effectiveness of these frameworks, rigorous experimental protocols and baseline assessments were developed, ensuring reliability, reproducibility, and generalizability.

### 5.2.1 Evaluation Protocol

Table 2a presents the distribution of available labels in the clean set. This part of the CatScreen dataset was partitioned into training, validation, and testing subsets following a 70:10:20 split. To maintain data integrity and avoid data leakage, partitioning was performed strictly at the patient level, ensuring all images from a single patient reside exclusively within one subset. Furthermore, balanced representation of diagnostic categories across all partitions was carefully maintained, reducing biases and ensuring robust model generalization across varied diagnostic scenarios.

### 5.2.2 Baseline Algorithms

Benchmark experiments were conducted using both Structured Sequential Analysis and Multitask Learning frameworks to establish foundational performance metrics. The baseline evaluations were performed under both unimodal and multimodal settings, thereby demonstrating the contribution and effectiveness of incorporating metadata.

**Structured Sequential Analysis Baseline**: In this setup, each diagnostic attribute — including image quality (Q), illumination type (T), overall diagnosis (D), cataract subtype (C), and severity grade (G) is predicted independently using separate models, without explicitly cascading outputs from one stage to another. Instead of enforcing explicit sequential conditioning, each prediction implicitly captures relevant cues from the image that may be shared across attributes. For example, although severity and subtype are clinically related, no direct sequential dependencies are imposed during training; each model learns to internally extract latent information from the image that implicitly reflects these relationships. This design serves as a fundamental baseline to understand the inherent capacity of visual representations to support each task in isolation while still allowing the model to leverage potential cross-attribute signals implicitly embedded in the learned features.

To demonstrate the applicability of this framework, we implement a conditional sequential inference pipeline at the evaluation stage, combining independently trained models to mimic the clinical diagnostic workflow. In this setup, the predictions from earlier attributes are conditionally used to guide subsequent tasks. This enables a structured decision flow without retraining the individual models. It helps to assess both the intrinsic discriminative capacity of visual representations in isolation and the effectiveness of structured sequential inference when independent predictions are composed in a clinically meaningful order.

**Multitask Learning Baseline**: In contrast, the multitask baseline jointly predicts all labels in a single end-to-end model. A common feature extractor first learns an overall representation from the slit-lamp image. Baseline experiments are performed in both unimodal and multimodal settings. separate prediction heads work in parallel to estimate image quality (Q), illumination type (T), diagnosis (D), cataract subtype (C), and severity grade (G). The model is trained using a combined loss that balances each task, helping it learn shared information across tasks while allowing each prediction head to focus on its specific details. This formulation enables the model to exploit inter-task correlations more explicitly within a single unified architecture and to quantify the impact of metadata inclusion, which supports rapid multi-attribute inference in a clinical setting.

For a comprehensive assessment, diverse deep learning architectures were employed, encompassing convolutional neural networks (CNNs) and transformer-based models. Specifically, models such as VGG-16 (Simonyan & Zisserman, 2015), ResNet18 (He et al., 2016), ResNet34 (He et al., 2016), MobileNet-V2 (Sandler et al., 2018), EfficientNet-B4 (Tan & Le, 2019), DenseNet-121 (Huang et al., 2017), swin transformer (swin_base_patch4_window_224) (Liu et al., 2021), and vision transformer (vit_base_patch16_224) (Dosovitskiy et al., 2021), all pretrained on the ImageNet dataset (Deng et al., 2009), were evaluated to benchmark model performance across different architectural paradigms. These were selected for their proven performance in image classification, balancing efficiency and feature extraction.

### 5.2.3 Implementation Details

All models were trained for 50 epochs using the AdamW optimizer with an initial learning rate of 0.0001 and a weight decay of $1 \times 10^{-5}$. Weighted cross-entropy loss was employed to mitigate class imbalances

prevalent within ophthalmic datasets. Computational experiments were executed using PyTorch on a DGX workstation equipped with 256 GB RAM and four Nvidia V100 GPUs (each with 32 GB memory).

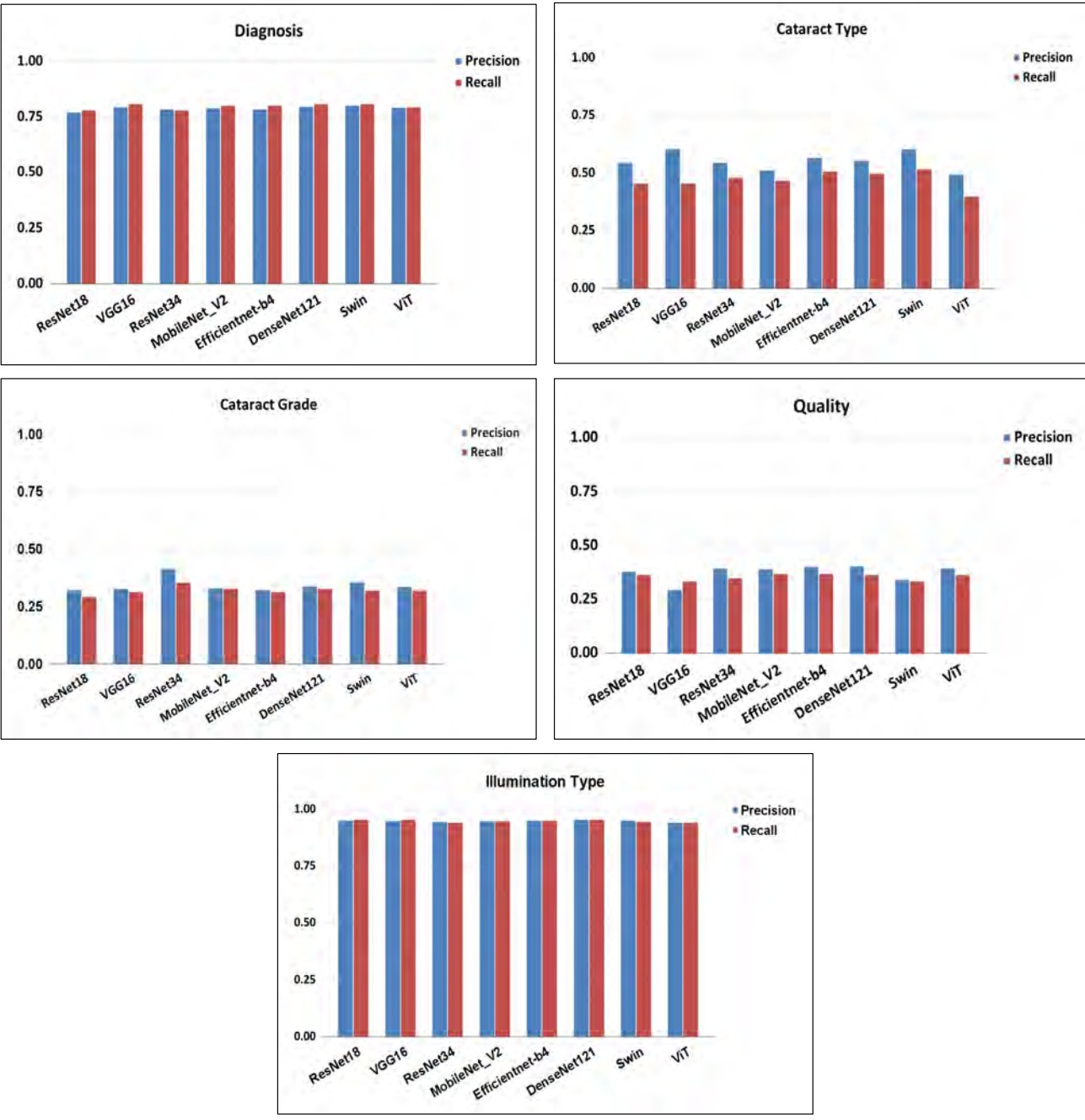

Figure 5: Showcasing the mean precision and recall scores across different diagnostic tasks in a unimodal setting. The plots demonstrate how model performance varies with task complexity, highlighting good performance for simpler and balanced classes such as illumination and comparatively lower precision-recall trade-offs for fine-grained prediction tasks.

## 6 Results and Analysis

This section analyzes the baseline performance outcomes obtained using the Independent Label Prediction, Structured Sequential Analysis and Multitask Learning frameworks. The performance metrics used to evaluate these approaches include the F1 score, precision, and recall, providing an insight into each framework's strengths and limitations. In addition to quantitative evaluation, this section examines the influence of image quality on cataract diagnosis through structured interpretability analyses. Qualitative visualization results are also provided to further strengthen the effectiveness of the proposed framework. Finally, this section discusses key technical challenges identified during experimentation and explains how the CatScreen dataset effectively mitigates these issues.

| Models/Classes | Res18 | VGG-16 | Res34 | M_V2 | E-b4 | DN121 | Swin | ViT |
|---|---|---|---|---|---|---|---|---|
| **Quality** | | | | | | | | |
| Good | 0.00 | 0.00 | 0.00 | 0.00 | 0.07 | 0.00 | 0.00 | 0.07 |
| Acceptable | 0.92 | 0.93 | 0.93 | 0.92 | 0.91 | 0.92 | 0.92 | 0.91 |
| Poor | 0.12 | 0.00 | 0.12 | 0.19 | 0.13 | 0.18 | 0.05 | 0.12 |
| **Illumination Type** | | | | | | | | |
| Diffuse | 0.98 | 0.98 | 0.98 | 0.98 | 0.98 | 0.98 | 0.98 | 0.98 |
| Direct Focal | 0.94 | 0.94 | 0.93 | 0.94 | 0.94 | 0.95 | 0.94 | 0.93 |
| Retro | 0.92 | 0.92 | 0.91 | 0.92 | 0.92 | 0.93 | 0.91 | 0.90 |
| **Diagnosis** | | | | | | | | |
| Normal | 0.60 | 0.66 | 0.60 | 0.64 | 0.64 | 0.65 | 0.65 | 0.64 |
| Cataract | 0.78 | 0.79 | 0.80 | 0.79 | 0.79 | 0.80 | 0.81 | 0.81 |
| Others | 0.93 | 0.94 | 0.95 | 0.94 | 0.94 | 0.95 | 0.94 | 0.92 |
| **Cataract Type** | | | | | | | | |
| NA | 0.61 | 0.62 | 0.67 | 0.64 | 0.62 | 0.63 | 0.67 | 0.58 |
| Nuclear | 0.74 | 0.74 | 0.77 | 0.75 | 0.74 | 0.75 | 0.75 | 0.72 |
| Cortical | 0.36 | 0.30 | 0.38 | 0.38 | 0.45 | 0.52 | 0.40 | 0.13 |
| PSC | 0.13 | 0.08 | 0.16 | 0.11 | 0.25 | 0.13 | 0.32 | 0.00 |
| PP | 0.91 | 0.92 | 0.92 | 0.91 | 0.90 | 0.92 | 0.93 | 0.88 |
| Others | 0.11 | 0.21 | 0.10 | 0.11 | 0.21 | 0.11 | 0.17 | 0.13 |
| **Cataract Grade** | | | | | | | | |
| NA | 0.77 | 0.80 | 0.76 | 0.76 | 0.77 | 0.79 | 0.79 | 0.80 |
| Mild | 0.65 | 0.69 | 0.66 | 0.66 | 0.67 | 0.67 | 0.68 | 0.70 |
| Severe | 0.23 | 0.20 | 0.21 | 0.21 | 0.19 | 0.19 | 0.28 | 0.15 |

Table 3: F1 Score comparison of independent label prediction across various backbones under a unimodal setting. (Here Res: ResNet, M_V2: MobileNet-V2, E-b4: EfficientNet-B4, DN121: DenseNet121, PP: Pseudophakia, PSC: Posterior Subcapsular Cataract, NA: Not Applicable)

### 6.1 Independent Label Prediction under unimodal and multimodal settings

This section presents the F1 scores for each diagnostic task within the unimodal and multimodal settings. The metadata is combined with the clinical tasks, such as cataract diagnostic classification and severity prediction. Several tasks exhibit near-zero F1 scores across models. This behaviour arises from extreme class imbalance, where some classes represent only a small fraction of the dataset. These outcomes should be interpreted as reflecting data scarcity rather than an inherent inability of the models to learn discriminative features. Table 3 summarizes the F1 scores for each diagnostic attribute across various popular deep learning backbone architectures utilized for medical imaging classification tasks. Complementing these results, Figure 5 visually illustrates the mean precision and recall achieved by each model, facilitating intuitive comparisons. The results consistently indicate an inverse correlation between the complexity of the classification task and model performance. For simpler classification tasks, such as illumination type classification, nearly all models achieved excellent F1 scores (0.90–0.98). Particularly, the *Diffuse* and *Direct Focal* illumination types were classified with very high confidence, reflecting their distinctive visual characteristics. However, the *Retro*

| Models / Classes | Res18 | VGG-16 | Res34 | M_V2 | E-b4 | DN121 | Swin | ViT |
|---|---|---|---|---|---|---|---|---|
| **Diagnosis** | | | | | | | | |
| Normal | 0.67 | 0.67 | 0.66 | 0.63 | 0.68 | 0.67 | 0.69 | 0.66 |
| Cataract | 0.78 | 0.81 | 0.78 | 0.78 | 0.82 | 0.80 | 0.80 | 0.81 |
| Others | 0.94 | 0.97 | 0.94 | 0.95 | 0.96 | 0.96 | 0.95 | 0.94 |
| **Cataract Type** | | | | | | | | |
| NA | 0.65 | 0.64 | 0.66 | 0.61 | 0.62 | 0.66 | 0.62 | 0.65 |
| Nuclear | 0.73 | 0.74 | 0.74 | 0.75 | 0.75 | 0.72 | 0.41 | 0.70 |
| Cortical | 0.22 | 0.33 | 0.38 | 0.34 | 0.52 | 0.43 | 0.09 | 0.09 |
| PSC | 0.18 | 0.26 | 0.12 | 0.14 | 0.31 | 0.33 | 0.06 | 0.11 |
| PP | 0.92 | 0.94 | 0.92 | 0.93 | 0.94 | 0.94 | 0.74 | 0.89 |
| Others | 0.35 | 0.30 | 0.33 | 0.30 | 0.15 | 0.24 | 0.13 | 0.37 |
| **Cataract Grade** | | | | | | | | |
| NA | 0.77 | 0.80 | 0.78 | 0.76 | 0.80 | 0.78 | 0.79 | 0.77 |
| Mild | 0.63 | 0.68 | 0.61 | 0.67 | 0.65 | 0.66 | 0.71 | 0.62 |
| Severe | 0.27 | 0.27 | 0.27 | 0.25 | 0.33 | 0.26 | 0.25 | 0.30 |

Table 4: F1 Score comparison of independent label prediction across various backbones under multimodal setting (Here Res: ResNet, M_V2: MobileNet-V2, E-b4: EfficientNet-B4, DN121: DenseNet121, PP: Pseudophakia, PSC: Posterior Subcapsular Cataract, NA: Not Applicable)

illumination type showed marginally lower performance (0.90–0.93), possibly attributable to subtler visual cues and its similarity to direct focal illumination. Accordingly, results for severely under-represented classes should be interpreted as preliminary baselines rather than stable estimates of achievable task performance.

Table 4 summarizes the F1 scores for the individual tasks under the multimodal setting. These results establish a baseline for multimodal learning across individual tasks. Compared to an unimodal setting, these models show improved performance in diagnostic classification and severity grading across most backbones. This indicates that patient-level contextual information complements visual features in the diagnostic tasks. These results highlight the complementary role of metadata in improving the robustness of the clinical relevance of the proposed CatScreen dataset.

### 6.1.1 Image Quality Assessment

In the task of image quality assessment, the *Acceptable* class consistently achieved robust F1 scores ranging between 0.91 and 0.93. Conversely, *Good* and *Poor* categories showed significantly weaker performance, often with near-zero scores. This result highlights a notable data imbalance favoring the *Acceptable* class, emphasizing the need for data augmentation strategies or class-balanced loss functions to enhance model sensitivity towards minority classes.

### 6.1.2 Diagnostic Classification

Diagnostic classification demonstrated moderate-to-high accuracy across most models. The *Cataract* class notably exhibited high F1 scores (0.78–0.81), attributable to its distinct visual features and adequate representation in the dataset. In contrast, the *Normal* class showed relatively lower scores (0.60–0.66), reflecting the challenges of distinguishing early-stage cataracts from healthy eyes due to subtle visual overlaps. The *Others* category demonstrated strong performance (0.92–0.95) despite minor variations, suggesting some degree of heterogeneity in these conditions and highlighting opportunities for further data enrichment.

The multimodal models show an improved performance across all classes. The *Normal* class shows an increase in the F1 score from 0.60–0.66 in the unimodal setting to 0.63–0.69 with metadata. The *Cataract* and *Others* class also maintains strong performance. These results highlight an increase in diagnostic sensitivity without degrading performance.

| Models/Classes | Res18 | VGG-16 | Res34 | M_V2 | E-b4 | DN121 | Swin | ViT |
|---|---|---|---|---|---|---|---|---|
| **Image Quality** | | | | | | | | |
| Good | 0.07 | 0.05 | 0.00 | 0.00 | 0.00 | 0.00 | 0.00 | 0.02 |
| Acceptable | 0.85 | 0.90 | 0.88 | 0.87 | 0.80 | 0.88 | 0.90 | 0.90 |
| Poor | 0.24 | 0.16 | 0.22 | 0.19 | 0.19 | 0.18 | 0.11 | 0.12 |
| **Diagnosis** | | | | | | | | |
| Normal | 0.68 | 0.66 | 0.64 | 0.63 | 0.65 | 0.67 | 0.68 | 0.66 |
| Cataract | 0.73 | 0.70 | 0.73 | 0.74 | 0.76 | 0.71 | 0.79 | 0.76 |
| Others | 0.93 | 0.94 | 0.94 | 0.93 | 0.92 | 0.94 | 0.93 | 0.94 |
| **Cataract Type** | | | | | | | | |
| NA | 0.29 | 0.26 | 0.29 | 0.36 | 0.34 | 0.33 | 0.30 | 0.38 |
| Nuclear | 0.74 | 0.75 | 0.41 | 0.53 | 0.62 | 0.69 | 0.27 | 0.68 |
| Cortical | 0.33 | 0.31 | 0.29 | 0.30 | 0.32 | 0.26 | 0.14 | 0.09 |
| PSC | 0.04 | 0.24 | 0.10 | 0.17 | 0.09 | 0.25 | 0.09 | 0.08 |
| PP | 0.08 | 0.67 | 0.06 | 0.16 | 0.23 | 0.12 | 0.06 | 0.07 |
| Others | 0.00 | 0.00 | 0.00 | 0.10 | 0.00 | 0.00 | 0.00 | 0.00 |
| **Cataract Grade** | | | | | | | | |
| NA | 0.27 | 0.19 | 0.36 | 0.33 | 0.38 | 0.24 | 0.18 | 0.33 |
| Mild | 0.62 | 0.77 | 0.73 | 0.74 | 0.67 | 0.69 | 0.65 | 0.69 |
| Severe | 0.31 | 0.19 | 0.29 | 0.28 | 0.37 | 0.29 | 0.31 | 0.30 |

Table 5: F1-scores for the multimodal Structured Sequential Analysis (SSA) pipeline across backbones for clinical tasks. (Here Res: ResNet, M_V2: MobileNet-V2, E-b4: EfficientNet-B4, DN121: DenseNet121, PP: Pseudophakia, PSC: Posterior Subcapsular Cataract, NA: Not Applicable)

### 6.1.3 Cataract Type Classification

Cataract subtype classification proved significantly challenging due to subtle visual distinctions and class imbalance. *Pseudophakia (PP)* achieved consistently high F1 scores (0.88–0.93) due to its visually distinct characteristics. However, subtypes like *Posterior Subcapsular Cataract (PSC)* and *Cortical* performed poorly (F1 scores ranging from 0.00–0.32 and 0.13–0.52 respectively), highlighting difficulties arising from their subtle visual patterns and limited training data. The *Nuclear* subtype showed intermediate to strong performance (0.72–0.77), benefiting from central opacity features, while the *Others* subtype presented highly inconsistent results (0.10–0.21), primarily driven by data scarcity.

Cataract subtype classification, such as Pseudophakia and Nuclear Cataracts, maintains good performance in both unimodal and multimodal settings. However, predicting more challenging cases, such as *PSC* and *Cortical* classes, shows good improvements in the multimodal setting. Due to visual ambiguity and class imbalance, fine-grained subtype classification remains challenging and will require more robust models that handle such limitations in future work. In particular, the results for *PSC* and *Others* subtypes should be viewed primarily as reference points for future class-imbalance-aware methods.

### 6.1.4 Cataract Severity Grading

Cataract severity grading emerged as particularly challenging, with *Severe* cases consistently showing poor F1 scores (0.15–0.28), reflecting significant class imbalance and limited representation. The *Not Applicable (NA)* category, representing non-cataract cases, achieved higher scores (0.76–0.80), and the *Mild* class produced moderate scores (0.65–0.70). These results emphasize the need for richer datasets that include adequate representation of severe cataract cases and advanced model architectures to accurately capture severe pathological patterns.

The impact of metadata is more pronounced in predicting cataract severity grading. Compared to the unimodal baseline, the multimodal framework achieves higher F1 scores for the *Severe* class, increasing from 0.15–0.28 in the unimodal setting to 0.25–0.33 in the multimodal setting. This improvement highlights that patient-level contextual information complements visual features and that severity grading is influenced by

| EfficientNet-B4 | | SSA | | Direct | |
|---|---|---|---|---|---|
| | | Acc | Macro F1 | Acc | Macro F1 |
| SSA Poor | Quality | 0.22 | 0.12 | 0.22 | 0.12 |
| | Diagnosis | - | - | 0.83 | 0.81 |
| | Grade | - | - | 0.57 | 0.38 |
| SSA Acceptable | Quality | 0.87 | 0.31 | 0.87 | 0.31 |
| | Diagnosis | 0.78 | 0.81 | 0.78 | 0.81 |
| | Grade | 0.59 | 0.46 | 0.54 | 0.34 |
| GT Poor | Quality | 0.25 | 0.13 | 0.25 | 0.13 |
| | Diagnosis | 0.79 | 0.68 | 0.79 | 0.66 |
| | Grade | 0.56 | 0.40 | 0.49 | 0.32 |
| GT Acceptable | Quality | 0.86 | 0.31 | 0.86 | 0.31 |
| | Diagnosis | 0.78 | 0.81 | 0.79 | 0.81 |
| | Grade | 0.59 | 0.46 | 0.55 | 0.35 |

Table 6: Impact of image quality on quality assessment, cataract diagnosis, and severity grading using EfficientNet-B4 under SSA and Individual label prediction (Direct) settings.

factors beyond visual appearance. Nevertheless, due to class imbalance, the overall performance gains remain constrained. This underscores the need for more balanced and class-aware learning strategies to build robust models.

## 6.2 SSA Framework with Structured Evaluation and Post-hoc Interpretability Analysis

To examine the behavior of the SSA formulation in a clinically structured inference setting, we implement a conditional sequential inference pipeline that combines independently trained single-task models to mimic the realistic diagnostic workflow. During training, the best model for each task is selected based on the F1 Score. Experimental results are presented in Table 5. The pipeline operates as follows: image quality is assessed, and poor-quality images are rejected for further processing. Diagnostic classification is subsequently performed using both image and metadata inputs. Images classified as normal or as others are excluded from the severity assessment. This conditional execution reflects real-world cataract screening, where early stages determine whether further diagnostic analysis is required.

Since the number of images passing the quality gate varies across backbone architectures, we evaluate the SSA framework using test sets consisting of SSA-Acceptable and SSA-Poor images. This set comprises images that pass the quality gate or are rejected, respectively, in the pipeline. To maintain clarity, we have presented results on the best-performing model, EfficientNet-B4.

Table 6 presents the impact of quality gate on subsequent task evaluations. Initial SSA-based evaluations revealed that performance differences between SSA-Poor and SSA-Acceptable subsets were not always consistently degradative across downstream tasks. This is mainly due to the severe class imbalance in image quality labels, which limits the reliability of the quality assessment model in certain cases. As a result, there is a gap between predicted and annotator-provided quality labels. To validate the underlying clinical hypothesis that poor-quality images degrade diagnostic and grading performance, we repeated the analysis using ground-truth quality labels. This ground-truth-based evaluation demonstrates a consistent and expected degradation in both cataract diagnosis and severity grading performance for poor-quality images.

To provide additional post-hoc insight into model behavior, we conducted GradCAM analyses for image-based predictions and SHAP analyses for metadata contributions. Figure 6a presents the GradCAM for the correctly classified and misclassified samples. The correctly classified samples show that the models focus on relevant lens regions, whereas misclassified cases exhibit diffuse or misplaced attention patterns that focus on peripheral or non-diagnostic regions. In parallel, Figure 6b presents the top features that influence the cataract diagnosis and severity grading tasks. The SHAP-based analysis elucidates that diagnostic classification relies on surgical history-related features. This reflects the strong association with post-operative lens status and non-cataract conditions. In contrast, cataract severity grading exhibits greater sensitivity

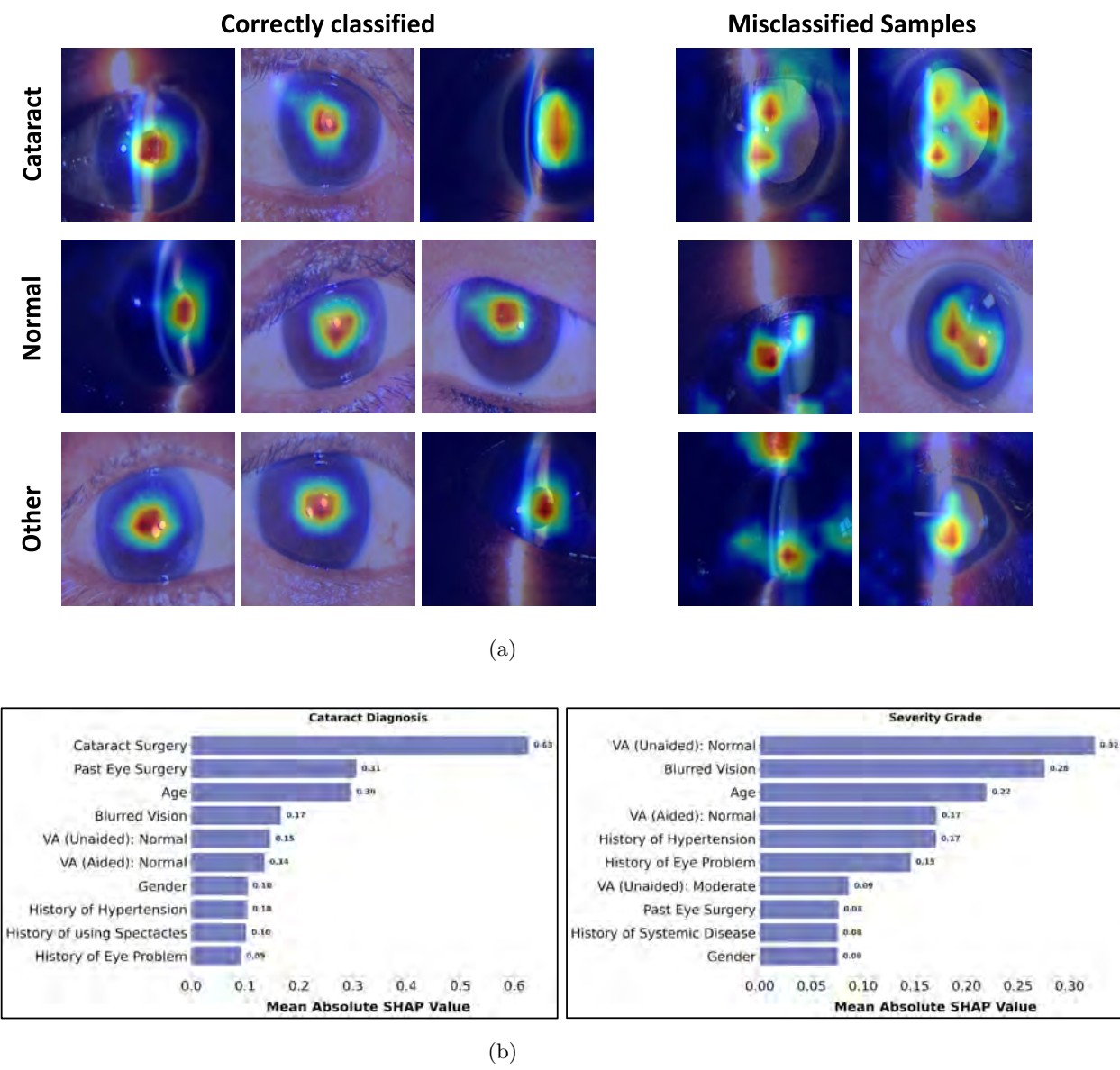

(a)

(b)

Figure 6: Presents (a) GradCAM visualizations for correctly classified and misclassified samples in the multimodal SSA framework using EfficientNet-B4 and (b) SHAP value plots highlighting the relative importance of patient-level metadata features for cataract diagnosis and severity grading.

to functional vision metrics, such as unaided visual acuity, blurred vision, and age. This observation aligns with clinical practice, in which visual function is a key indicator of cataract disease progression. These analyses provide qualitative, post-hoc evidence about model attention and feature importance, but they should not be interpreted as definitive validation of the model's reasoning process or as formal proof of superior interpretability of the SSA formulation.

## 6.3 Multitask Learning Framework

Tables 7 and 8 summarize the F1 scores for each diagnostic attribute within the multitask learning framework, where multiple tasks are predicted simultaneously from shared visual representations in unimodal and multimodal settings, respectively. To assess the impact of metadata within the multitask learning paradigm,

| Models/Classes | Res18 | VGG-16 | Res34 | M_V2 | E-b4 | DN121 | Swin | ViT |
|---|---|---|---|---|---|---|---|---|
| **Quality** | | | | | | | | |
| Good | 0.00 | 0.00 | 0.00 | 0.00 | 0.00 | 0.00 | 0.00 | 0.00 |
| Acceptable | 0.93 | 0.93 | 0.93 | 0.92 | 0.93 | 0.93 | 0.93 | 0.93 |
| Poor | 0.00 | 0.00 | 0.00 | 0.00 | 0.00 | 0.00 | 0.00 | 0.10 |
| **Illumination Type** | | | | | | | | |
| Diffuse | 0.98 | 0.98 | 0.98 | 0.98 | 0.98 | 0.98 | 0.98 | 0.98 |
| Direct Focal | 0.95 | 0.94 | 0.93 | 0.94 | 0.93 | 0.94 | 0.94 | 0.94 |
| Retro | 0.92 | 0.92 | 0.92 | 0.92 | 0.90 | 0.91 | 0.92 | 0.91 |
| **Diagnosis** | | | | | | | | |
| Normal | 0.63 | 0.66 | 0.64 | 0.63 | 0.60 | 0.65 | 0.67 | 0.66 |
| Cataract | 0.79 | 0.81 | 0.80 | 0.80 | 0.79 | 0.80 | 0.80 | 0.82 |
| Others | 0.91 | 0.91 | 0.92 | 0.89 | 0.91 | 0.92 | 0.94 | 0.92 |
| **Cataract Type** | | | | | | | | |
| NA | 0.65 | 0.66 | 0.63 | 0.64 | 0.61 | 0.65 | 0.67 | 0.67 |
| Nuclear | 0.76 | 0.75 | 0.76 | 0.76 | 0.74 | 0.76 | 0.75 | 0.78 |
| Cortical | 0.00 | 0.00 | 0.00 | 0.00 | 0.00 | 0.00 | 0.00 | 0.50 |
| PSC | 0.00 | 0.00 | 0.00 | 0.00 | 0.00 | 0.00 | 0.00 | 0.00 |
| PP | 0.90 | 0.92 | 0.89 | 0.87 | 0.89 | 0.91 | 0.92 | 0.91 |
| Others | 0.00 | 0.00 | 0.00 | 0.00 | 0.00 | 0.00 | 0.00 | 0.00 |
| **Cataract Grade** | | | | | | | | |
| NA | 0.79 | 0.80 | 0.78 | 0.78 | 0.78 | 0.79 | 0.81 | 0.81 |
| Mild | 0.69 | 0.71 | 0.69 | 0.71 | 0.70 | 0.70 | 0.71 | 0.74 |
| Severe | 0.00 | 0.00 | 0.00 | 0.00 | 0.00 | 0.00 | 0.00 | 0.09 |

Table 7: F1 Score comparison of multiple labels across various backbones as feature extractors for multitask learning approach. (Here Res: ResNet, M_V2: MobileNet-V2, E-b4: EfficientNet-B4, DN121: DenseNet121, PP: Pseudophakia, PSC: Posterior Subcapsular Cataract, NA: Not Applicable)

we also performed experiments for a multimodal multitask setting. We included metadata for clinically relevant tasks, and other tasks remain image-only, reflecting their dependence on acquisition conditions rather than patient context. Experimental results demonstrate increased performance gains in multimodal settings.

### 6.3.1 Image Quality Assessment and Illumination Type Classification

In multitask learning, image quality prediction again showed robust performance in the *Acceptable* category (F1: 0.92–0.93). Conversely, the *Good* and *Poor* categories exhibited critically low performance (F1 near 0), highlighting persistent challenges due to class imbalance. Interestingly, Vision Transformer (ViT) demonstrated slight improvement in the *Poor* category (F1: 0.10), emphasizing the potential of transformer architectures for capturing subtle visual differences. Further, illumination type classification demonstrated exceptional performance across all models, achieving nearly perfect scores for the *Diffuse* category (F1: 0.98). Strong performances were also recorded for *Direct Focal* and *Retro* illumination classes, with F1 scores ranging from 0.90 to 0.95, highlighting the consistency and reliability of models in handling clearly distinguishable visual features.

### 6.3.2 Diagnostic Classification and Cataract Type Classification

Diagnostic classification within the multitask framework again yielded strong outcomes for the *Cataract* (F1: 0.79–0.82) and *Others* (F1: 0.91–0.94) classes, reflecting their visually distinct and adequately represented nature. The *Normal* category continued to pose challenges (F1: 0.60–0.67), reiterating difficulties in differentiating healthy from early pathological states due to subtle visual overlaps. Next, subtype classification presented considerable challenges within the multitask framework. The *Pseudophakia (PP)* and *Nuclear* cataract classes showed commendable performances (F1: 0.87–0.92 and 0.74–0.78, respectively). In contrast, the *Cortical*, *PSC*, and *Others* subtypes consistently yielded near-zero scores, emphasizing the limitations

| Models/Classes | Res18 | VGG-16 | Res34 | M_V2 | E-b4 | DN121 | Swin | ViT |
|---|---|---|---|---|---|---|---|---|
| **Quality** | | | | | | | | |
| Good | 0.00 | 0.00 | 0.09 | 0.02 | 0.00 | 0.07 | 0.00 | 0.05 |
| Acceptable | 0.89 | 0.89 | 0.89 | 0.87 | 0.88 | 0.88 | 0.88 | 0.88 |
| Poor | 0.28 | 0.20 | 0.25 | 0.24 | 0.25 | 0.22 | 0.24 | 0.22 |
| **Illumination Type** | | | | | | | | |
| Diffuse | 0.98 | 0.98 | 0.98 | 0.98 | 0.98 | 0.98 | 0.98 | 0.98 |
| Direct Focal | 0.93 | 0.94 | 0.94 | 0.93 | 0.94 | 0.93 | 0.94 | 0.93 |
| Retro | 0.90 | 0.92 | 0.91 | 0.91 | 0.92 | 0.91 | 0.91 | 0.90 |
| **Diagnosis** | | | | | | | | |
| Normal | 0.65 | 0.67 | 0.64 | 0.63 | 0.66 | 0.68 | 0.67 | 0.64 |
| Cataract | 0.79 | 0.80 | 0.78 | 0.78 | 0.77 | 0.80 | 0.77 | 0.78 |
| Others | 0.91 | 0.96 | 0.93 | 0.94 | 0.95 | 0.96 | 0.95 | 0.92 |
| **Cataract Type** | | | | | | | | |
| NA | 0.67 | 0.65 | 0.65 | 0.64 | 0.65 | 0.67 | 0.64 | 0.62 |
| Nuclear | 0.70 | 0.74 | 0.71 | 0.70 | 0.71 | 0.72 | 0.75 | 0.69 |
| Cortical | 0.38 | 0.34 | 0.30 | 0.34 | 0.33 | 0.34 | 0.41 | 0.16 |
| PSC | 0.09 | 0.20 | 0.00 | 0.23 | 0.24 | 0.29 | 0.20 | 0.00 |
| PP | 0.91 | 0.94 | 0.91 | 0.93 | 0.93 | 0.93 | 0.94 | 0.91 |
| Others | 0.09 | 0.24 | 0.22 | 0.36 | 0.11 | 0.17 | 0.31 | 0.22 |
| **Cataract Grade** | | | | | | | | |
| NA | 0.79 | 0.78 | 0.77 | 0.76 | 0.79 | 0.79 | 0.79 | 0.78 |
| Mild | 0.63 | 0.67 | 0.61 | 0.61 | 0.66 | 0.69 | 0.67 | 0.60 |
| Severe | 0.23 | 0.20 | 0.21 | 0.28 | 0.20 | 0.21 | 0.34 | 0.30 |

Table 8: F1 Score comparison of multiple labels across various backbones as feature extractors for multimodal multitask learning approach. (Here Res: ResNet, M_V2: MobileNet-V2, E-b4: EfficientNet-B4, DN121: DenseNet121, PP: Pseudophakia, PSC: Posterior Subcapsular Cataract, NA: Not Applicable)

imposed by subtle visual differences, insufficient data, and high intra-class variability. The ViT architecture, however, demonstrated notable improvement for the *Cortical* subtype (F1: 0.50), confirming the potential of transformer-based models in fine-grained classification tasks.

The multimodal multitask framework demonstrates a consistent improvement over the unimodal baseline. The F1 score of the *Normal* class increases from approximately 0.60–0.67 in the unimodal setting to 0.63–0.68 with metadata across most backbones. For cataract type classification, there is a similar trend of consistent performance improvement across classes. F1 scores of cataract subtypes, such as *Cortical*, improve from near-zero values in the unimodal multitask setting to approximately 0.30–0.41 across several backbones, while PSC improves from near-zero to approximately 0.20–0.29 under the multimodal multitask framework. The performance of other classes remains stable or slightly improves, indicating that metadata serves as complementary information to visual information without affecting the prediction of visually distinctive classes.

### 6.3.3 Cataract Severity Grading

Severity grading results mirrored challenges observed in the sequential analysis. The *Not Applicable (NA)* and *Mild* classes performed reasonably well (F1: 0.70–0.81), while the *Severe* category remained problematic with consistently poor scores (near-zero F1 scores). The slight improvement observed in the *Severe* category by ViT (F1: 0.09) further supports transformer architectures' capability to manage complex visual variations better than traditional models. The inclusion of metadata helps to improve the performance of this task. Compared to the unimodal multitask baseline, the multimodal multitask framework improves the F1 score for the *Severe* class from near-zero values ($\geq 0.09$) to approximately 0.20–0.34 across different backbone architectures. These results demonstrate the impact of metadata on the performance of overall diagnostic tasks.

### 6.4 Future Research Directions

CatScreen's design explicitly supports research beyond supervised classification. While the present work focuses on baseline supervised benchmarks on the clean set and selected multimodal experiments, CatScreen is intentionally designed to support a broader range of future research directions. The noisy-labeled subset enables benchmarking of label-noise robust algorithms under realistic annotation conditions. The unlabeled subset of 5,481 images facilitates self-supervised pretraining and semi-supervised learning approaches. The anatomical segmentation annotations support development of interpretable models with explicit anatomical grounding. These components collectively position CatScreen as a comprehensive resource for advancing robust, generalizable, and interpretable cataract screening solutions.

Specifically, the noisy-labeled subset can be used to evaluate methods such as co-teaching, confident learning, and noise-robust loss functions. The unlabeled subset enables exploration of self-supervised pretraining frameworks (e.g., SimCLR, DINO, MAE) that can improve feature representations for downstream tasks. Comprehensive evaluation of these components represents natural extensions of this work.

### 6.5 Technical Challenges and CatScreen's Contributions

Our findings identify several promising directions for future research to address current limitations. Leveraging self-supervised pretraining techniques can enhance models' abilities to extract meaningful features from unlabeled ophthalmic data. Moreover, integrating advanced attention mechanisms and multi-scale feature refinement strategies could significantly boost model sensitivity to subtle visual distinctions, critical for effective fine-grained classification. These methods can help models better discriminate between visually similar ophthalmic conditions, potentially improving performance on fine-grained and clinically relevant screening tasks.

The multitask learning framework specifically requires careful balancing between shared representations and task-specific fine-tuning to mitigate negative transfer and improve overall task performance. Implementing dynamic task-weighting or uncertainty-based loss re-weighting strategies could address these challenges effectively. Conversely, the structured sequential analysis framework would benefit from explicitly modeling inter-task dependencies. Figure 7 showcases the sample result images with true and predicted labels that are misclassified due to poor-quality images. In sequential causal models, early-stage errors cascade forward and cause misclassifications in subsequent tasks that depend on these intermediate predictions. This emphasizes the critical need for robust handling of quality variations to mitigate cascading errors in sequential decision models. Approaches such as cascaded conditioning, multi-stage refinement modules, or cross-task attention mechanisms could leverage shared clinical features while maintaining modularity and improving diagnostic accuracy.

A key limitation of the proposed benchmark is the severe class imbalance present in certain classes, which limits the interpretability of the performance metrics for the rare classes. While this imbalance mirrors real-world clinical prevalence, it limits the reliable assessment of the model behaviour for these classes. Future research will address class imbalance using techniques like targeted data collection, data augmentation, synthetic oversampling, or focal loss methods, which could significantly improve sensitivity to minority classes, particularly severe cataract cases and extreme image quality categories. Further research into advanced attention mechanisms, ensemble approaches, or hierarchical classification strategies specifically targeted toward challenging cataract subtypes, such as Cortical and PSC, could better capture nuanced visual distinctions. Lastly, explicitly evaluating model robustness through the noisy-labeled subset by leveraging semi-supervised and robust learning approaches would enhance real-world generalizability and reliability.

To support these research directions effectively and address identified technical challenges, CatScreen provides several key contributions and innovative features:

- **Class Imbalance and Minority Class Performance**: Both SSA and MTL frameworks exhibited significantly reduced performance on minority classes. CatScreen preserves realistic class distributions and therefore provides a useful benchmark for future work on minority-class performance and class-imbalance-aware learning.

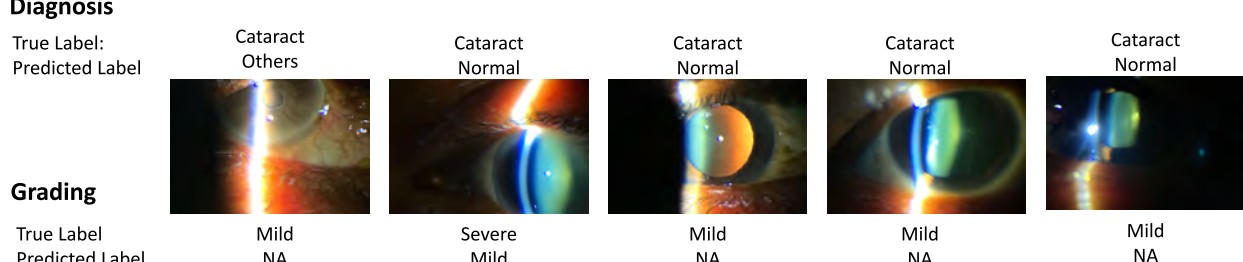

Figure 7: Presents the sample result images of poor-quality images causing misclassification across diagnosis and severity grade labels

- **Fine-grained Classification**: Differentiating subtle visual distinctions among cataract subtypes and severity levels remains challenging. CatScreen's fine-grained annotations and precise labeling enable models to learn nuanced, clinically relevant fine-grained distinctions, thereby supporting future work on subtype discrimination and anatomically grounded analysis.

- **Robustness to Annotation Noise**: Clinical datasets inherently contain annotation inaccuracies. CatScreen includes a Noisy Set specifically designed to develop and evaluate models robust against imperfect annotations, thus aligning more closely with real-world clinical practice.

- **Utilizing Unlabelled Data for Enhanced Representation Learning**: To boost generalization and robustness, CatScreen offers a substantial Unlabelled Set. This set supports the development of self-supervised, semi-supervised, and active learning techniques, enabling models to utilize extensive unlabeled data for improved feature extraction and stronger generalization.

## 7 Conclusion

CatScreen advances ophthalmic diagnostics by offering a richly annotated, large-scale slit-lamp image dataset tailored specifically for addressing real-world challenges in cataract screening. It captures a comprehensive range of clinically relevant annotations, including image gradability, illumination types, cataract subtypes, and severity grades, enabling the creation of highly interpretable and clinically meaningful AI solutions. Beyond traditional supervised learning, CatScreen dataset uniquely incorporates noisy-labelled and unlabeled data subsets, thereby facilitating research into robust self-supervised and semi-supervised learning strategies reflective of practical clinical scenarios. The detailed patient metadata further enhances its utility by supporting the integration of holistic clinical contexts into diagnostic models. In this work, we present baseline evaluations under independent, structured sequential, and multitask prediction settings in both unimodal and multimodal configurations. These experiments provide initial reference results and show that metadata can improve performance for selected diagnostic tasks, while also highlighting persistent challenges such as severe class imbalance, minority-class sensitivity, and fine-grained subtype discrimination. We anticipate that CatScreen will inspire extensive innovation, significantly accelerating the adoption of AI-driven eye care solutions and improving global cataract screening and management. Furthermore, the inclusion of noisy-labeled, unlabeled, and segmentation-annotated subsets provides researchers with resources to explore robust learning, semi-supervised methods, and anatomically-grounded approaches in future studies.

## 8 Broader Impact Considerations

As a medical imaging dataset intended to support cataract screening in resource-limited settings, several broader considerations warrant discussion. First, the class imbalance present in CatScreen, particularly the underrepresentation of severe cataract cases (9.6% of the dataset), may result in models with lower sensitivity for the most clinically urgent cases. Deployment strategies should account for this limitation through appropriate threshold calibration or ensemble approaches.

Second, while patient metadata enhances diagnostic performance, it also introduces potential for demographic biases. Models leveraging metadata should be evaluated across demographic subgroups (age, gender, diabetic status) to ensure equitable performance before clinical deployment.

Finally, CatScreen is designed to support screening and triage workflows rather than replace clinical judgment. Any real-world deployment should maintain appropriate human oversight by trained healthcare professionals, with AI-assisted predictions serving as decision support rather than autonomous diagnosis.

## Acknowledgment

This work was supported by iHub-Drishti, the Technology Innovation Hub at IIT Jodhpur.

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
