# OpenReview forum: "CatScreen: A Large MultiModal Benchmark Dataset for Cataract Screening"
_TMLR — Accepted by TMLR_

### Review · Reviewer_aGs3 · 2025-09-21

**Summary Of Contributions:**

# contribution
This paper presents CatScreen, a large-scale multimodal dataset for cataract screening. Its core contributions are: 1) providing a substantial dataset of ~18.6k images from 2.25k subjects, 2) partitioning the data into clean, noisy, and unlabeled subsets to support various learning paradigms, and 3) proposing and benchmarking two complementary AI frameworks (Structured Sequential Analysis and Multitask Learning) on the new dataset.

# strength
- The dataset is larger than many existing resources.
-  It features fine-grained, multi-dimensional labels and includes pixel-level anatomical segmentations and patient metadata.

# weakness
- The paper does not detail how the noisy labels in the noisy-labeled set were generated. Without knowing the source or the nature of the noise, it is difficult to assess how well this subset truly emulates real-world clinical noise and, consequently, its utility for developing robust learning algorithms.

- The experiments do not incorporate the collected metadata, leaving its actual benefit to the task unproven.

- The claimed interpretability advantage of the sequential framework is not demonstrated with sufficient evidence.

**Additional Comments:**

/

**Audience:**

Yes

**Audience Explanation:**

- The dataset is larger than many existing resources.
- The dataset provides fine-grained annotations including image quality, cataract subtype, severity, and rare pixel-level anatomical labels.

**Claims And Evidence:**

No

**Claims Explanation:**

Refer to the weakness part:
- The method for generating the noisy labels is not described, making its realism and utility unclear.
- The actual benefit of metadata to the task remains unproven.
- The claimed interpretability advantage of the sequential framework is not demonstrated with sufficient evidence.

**Requested Changes:**

As said in the weakness part, more detailed descriptions about 1)  how the noisy label set was generated, 2) the interpretability of the proposed framework, and 3) the empirical evidence about the benefit of metadata are needed.

---

> ### Author Response · Authors · 2026-01-20
> **We thank the reviewer for providing their feedback and suggestions. Our response to the reviewer’s questions are provided below**
>
> **Generation of noisy label set**: The CatScreen dataset was collected from one hospital and community-based settings. The annotation process was structured into three levels to ensure the reliability of the clean set:
> * Level 1: Initial annotations performed by experienced optometrists holding a bachelor's degree in optometry with a minimum of three years of hands-on experience in retinal and anterior segment imaging. These annotations provided a foundational layer for subsequent refinement.
> * Level 2: A secondary review performed by a senior optometrist with over five years of clinical and imaging experience. This level involved resolving any ambiguities and discrepancies by thoroughly refining and correcting initial annotations.
> * Level 3: A retina-trained ophthalmologist with over a decade of clinical and diagnostic experience provided final validation and oversight. This expert-level validation ensured maximum accuracy, consistency, and clinical relevance of the final annotations.
>
> Only the data that successfully passed all three levels, with discrepancies corrected and final verification by the ophthalmologist, were included in the clean set.
> In contrast, the noisy set consisted of cases annotated solely at Level 1 (by optometrists) without subsequent review by senior optometrists or ophthalmologists. Since Level 1 annotations may contain errors due to differences in expertise and diagnostic precision, this subset inherently incorporates real-world label noise, particularly representative of community-based screening contexts where specialist review is often unavailable. Therefore, the noisy-labeled subset of the CatScreen dataset reflects the type of clinical noise encountered in practical, resource-limited settings, making it essential for developing and benchmarking robust learning algorithms. We have added this clarification in the data collection section (Section 3.2) in the revised manuscript.
>
> **Impact of metadata**: We acknowledge the value of metadata in enhancing clinical AI systems. To address this, we conducted experiments in a multimodal setting that included both images and metadata for independent individual-label prediction, as well as the multitask and SSA framework. Since the metadata consisted of clinical features, we used it solely for cataract diagnosis and for predicting cataract type and grade.  Here, we used two encoders, one for each modality, and then concatenated them into a single feature vector before the final prediction. The experimental results are reported in Tables 4, 5, and 8 of the updated manuscript.

---

> ### Author Response · Authors · 2026-01-20
>
> **Interpretability of the sequential framework**: To demonstrate the interpretability of the structured sequential framework, we implement a conditional sequential inference pipeline that combines the independently trained single-task models to mimic the diagnostic workflow. During training, the best model for each task is selected based on the F1 Score. Experimental results are presented in Table 5 of the revised manuscript. The pipeline operates as follows: image quality is assessed, and poor-quality images are rejected for further processing. Diagnostic classification is subsequently performed using both image and metadata inputs. Images classified as normal or as others are excluded from the severity assessment. This conditional execution reflects real-world cataract screening, where early stages determine whether further diagnostic analysis is required.
>
> Since the number of images passing the quality gate varies across backbone architectures, we evaluate the SSA framework using test sets consisting of SSA-Acceptable and SSA-Poor images. This set comprises images that pass the quality gate or are rejected, respectively, in the pipeline. To maintain clarity, we have presented results on the best-performing model, EfficientNet-B4.
>
> Initial SSA-based evaluations revealed that performance differences between SSA-Poor and SSA-Acceptable subsets were not always consistently degradative across downstream tasks. This is mainly due to the severe class imbalance in image quality labels, which limits the reliability of the quality assessment model in certain cases. As a result, there is a gap between predicted and annotator-provided quality labels. To validate the underlying clinical hypothesis that poor-quality images degrade diagnostic and grading performance, we repeated the analysis using ground-truth quality labels. This ground-truth-based evaluation demonstrates consistent, expected degradation in both cataract diagnosis and severity grading performance for poor-quality images. The results are reported in Table 6 of the revised manuscript.
>
> To further strengthen the interpretability of the proposed SSA framework, we conducted post-hoc explainability analyses using GradCAM for image-based predictions and SHAP for metadata-driven contributions. Figure 6(a) presents the GradCAM for correctly and incorrectly classified samples. The correctly classified samples show that the models focus on relevant lens regions, whereas misclassified cases exhibit diffuse or misplaced attention patterns that focus on peripheral or non-diagnostic regions. In parallel, Figure 6(b) presents the top features that influence the cataract diagnosis and severity grading tasks. The SHAP-based analysis elucidates that diagnostic classification relies on surgical history-related features. This reflects the strong association with post-operative lens status and non-cataract conditions. In contrast, cataract severity grading exhibits greater sensitivity to functional vision metrics, such as unaided visual acuity. This observation aligns with clinical practice, in which visual function is a key indicator of disease progression.

---

### Review · Reviewer_Tvtv · 2025-12-15

**Summary Of Contributions:**

This paper introduces CatScreen, a large-scale, multimodal slit-lamp image dataset for cataract screening, comprising a large number of images captured using a low-cost portable device. Relative to prior cataract datasets, CatScreen is distinguished by its breadth of clinically relevant annotations per image, including image gradability and quality, illumination type, diagnostic category (normal / cataract / other), cataract subtype, and severity grading. In addition, the dataset includes subject-level demographic and clinical metadata, a noisy-labeled subset, an unlabeled subset, and segmentation masks for key ocular structures on a subset of images. The paper also presents two baseline modeling paradigms and benchmarks a range of CNN and transformer backbones. The primary contribution is therefore the dataset itself, positioned as a comprehensive benchmark for developing and evaluating clinically grounded cataract screening models, rather than new machine learning methodology.

**Audience:**

Yes

**Audience Explanation:**

This work should be of interest to researchers focused on medical imaging datasets, AI for ophthalmology, and dataset design for real-world clinical deployment, particularly in low-resource settings. Compared to existing cataract datasets, CatScreen offers a more comprehensive and realistic annotation schema that better reflects clinical workflows. However, the appeal is primarily to applied ML and medical imaging researchers rather than to those seeking new algorithmic or theoretical contributions in machine learning.

**Broader Impact Concerns:**

No major ethical concerns are apparent. The paper could consider briefly acknowledging standard considerations relevant to medical screening datasets, such as the implications of class imbalance (particularly for severe cases), potential demographic biases when using patient metadata, and the importance of human oversight in any real-world deployment.

**Claims And Evidence:**

No

**Claims Explanation:**

When contextualized against the existing literature, the dataset contribution is meaningful. Prior slit-lamp datasets typically involve on the order of hundreds of subjects or a few thousand images, often limited to a single cataract subtype or severity axis. In contrast, CatScreen’s scale, labeling, and subtype diversity present a richer data source.

The data collection protocol, ethical approvals, and three-tier annotation process are described in sufficient detail to support claims of annotation quality and clinical relevance. The benchmark experiments also consistently reflect known challenges in ophthalmic imaging, particularly severe class imbalance and the difficulty of fine-grained subtype and severity classification, lending credibility to the reported results.

However, several claims are stronger than what is empirically demonstrated. The paper emphasizes interpretability, causal structure, and robustness to real-world noise, but the experimental section does not convincingly validate these properties. The structured sequential analysis framework is motivated as causally informed, yet no quantitative or qualitative analysis demonstrates improved interpretability, reduced error propagation, or causal insight compared to standard multitask learning. Similarly, while noisy labels and unlabeled data are highlighted as key dataset components, they are not substantively used in the reported experiments. Thus, while the dataset’s potential is clear, some downstream claims remain aspirational rather than evidenced.

**Requested Changes:**

* While the structured sequential analysis framework is motivated as interpretable and causally aligned with clinical reasoning, the paper does not empirically demonstrate these benefits. At present, this framing remains largely conceptual. Either stronger evidence should be provided (e.g., error propagation analysis, clinician-aligned interpretability studies), or the claims should be softened.
* The dataset’s most novel components: metadata, noisy labels, unlabeled data, and segmentation masks, are largely unused in the experiments. As a result, the benchmarks do not fully justify the emphasis placed on these features in the paper. Incorporating even preliminary experiments using metadata (it is unclear whether the experiments for the structured sequential analysis framework incorporate this) or noisy-label robustness would substantially strengthen the contribution.
* Many tasks (e.g., quality “Good”/“Poor”, PSC subtype, severe cataract grading) yield near-zero F1 scores across almost all models. While this reflects real-world data distributions, the paper does not sufficiently discuss how such results should be interpreted or how future work might meaningfully address them. This severe class imbalance limits the interpretability of the benchmark.
* Figure 5 is very low resolution, visually cluttered, and difficult to interpret. Axis labels are barely legible, legends are unclear, and the dense layout obscures meaningful comparison across models and tasks. Given that this figure is central to the empirical evaluation, it should be substantially redesigned or replaced with clearer tables.
* Several figures (Figures 3, 5) could be simplified without loss of content. Additionally, these figures are very low resolution and hard to read. Some tables (e.g. table 3) duplicate information already described in the text.

---

> ### Author Response · Authors · 2026-01-20
> **We thank the reviewer for providing their feedback and suggestions. Our response to the reviewer’s questions are provided below:**
>
> **Interpretability of the sequential framework**: To demonstrate the interpretability of the structured sequential framework, we implement a conditional sequential inference pipeline that combines the independently trained single-task models to mimic the diagnostic workflow. During training, the best model for each task is selected based on the F1 Score. Experimental results are presented in Table 5 of the revised manuscript. The pipeline operates as follows: image quality is assessed, and poor-quality images are rejected for further processing. Diagnostic classification is subsequently performed using both image and metadata inputs. Images classified as normal or as others are excluded from the severity assessment. This conditional execution reflects real-world cataract screening, where early stages determine whether further diagnostic analysis is required.
>
> Since the number of images passing the quality gate varies across backbone architectures, we evaluate the SSA framework using test sets consisting of SSA-Acceptable and SSA-Poor images. This set comprises images that pass the quality gate or are rejected, respectively, in the pipeline. To maintain clarity, we have presented results on the best-performing model, EfficientNet-B4.
>
> Initial SSA-based evaluations revealed that performance differences between SSA-Poor and SSA-Acceptable subsets were not always consistently degradative across downstream tasks. This is mainly due to the severe class imbalance in image quality labels, which limits the reliability of the quality assessment model in certain cases. As a result, there is a gap between predicted and annotator-provided quality labels. To validate the underlying clinical hypothesis that poor-quality images degrade diagnostic and grading performance, we repeated the analysis using ground-truth quality labels. This ground-truth-based evaluation demonstrates consistent, expected degradation in both cataract diagnosis and severity grading performance for poor-quality images. The results are reported in Table 6 of the revised manuscript.
> To further strengthen the interpretability of the proposed SSA framework, we conducted post-hoc explainability analyses using GradCAM for image-based predictions and SHAP for metadata-driven contributions. Figure 6(a) presents the GradCAM for correctly and incorrectly classified samples. The correctly classified samples show that the models focus on relevant lens regions, whereas misclassified cases exhibit diffuse or misplaced attention patterns that focus on peripheral or non-diagnostic regions. In parallel, Figure 6(b) presents the top features that influence the cataract diagnosis and severity grading tasks. The SHAP-based analysis elucidates that diagnostic classification relies on surgical history-related features. This reflects the strong association with post-operative lens status and non-cataract conditions. In contrast, cataract severity grading exhibits greater sensitivity to functional vision metrics, such as unaided visual acuity. This observation aligns with clinical practice, in which visual function is a key indicator of disease progression.

---

> ### Author Response · Authors · 2026-01-20
>
> **Impact of metadata**: We acknowledge the value of metadata in enhancing clinical AI systems. To address this, we conducted experiments in a multimodal setting that included both images and metadata for independent individual-label prediction, as well as the multitask and SSA framework. Since the metadata consisted of clinical features, we used it solely for cataract diagnosis and for predicting cataract type and grade.  Here, we used two encoders, one for each modality, and then concatenated them into a single feature vector before the final prediction. The experimental results are reported in Tables 4, 5, and 8 of the updated manuscript.
>
> **Regarding noisy labels, unlabeled data, and segmentation masks**: We acknowledge that the current benchmark focuses on establishing supervised learning baselines with metadata integration. The noisy-labeled subset, unlabeled subset, and segmentation annotations are intentionally included as resources to enable diverse research directions beyond the scope of this initial benchmark. Specifically:
> * Noisy-labeled subset: Designed to support research in label-noise robust learning, enabling evaluation of methods such as co-teaching, MentorNet, confident learning, and other noise-robust approaches.
> * Unlabeled subset (5,481 images): Enables exploration of self-supervised pretraining (e.g., SimCLR, DINO, MAE) and semi-supervised learning frameworks that can leverage large amounts of unlabeled ophthalmic data.
> * Segmentation masks: Support interpretability studies, anatomically-grounded model development, and can be used to train or evaluate segmentation models for anterior segment structures.
>
> We have added a discussion in Section 6.4 explicitly outlining these research opportunities. Comprehensive evaluation of these components represents natural extensions of this work and will be addressed in subsequent studies. The current baselines with metadata integration (Tables 4, 5, and 8) demonstrate the dataset's utility and establish reference points for future algorithmic development.
>
> **Low F1 scores**: We acknowledge that several tasks contain classes that are extremely underrepresented in the dataset. The reviewer has correctly noted that many tasks yield near-zero or zero F1 scores, indicating data scarcity. Under such conditions, the F1 score, being sensitive to both precision and recall, tends to approach zero even when the model produces some correct predictions.
>
> These results therefore reflect the inherent data imbalance rather than a fundamental limitation of the modeling frameworks. Specifically:
> * "Good" quality: 176 samples (1.8% of clean set)
> * PSC subtype: 113 samples (1.1%)
> * Severe grade: 952 samples (9.6%)
>
> We have updated the manuscript to discuss these results in Section 6.4. Additionally, we have outlined future directions to address these limitations:
> * Targeted data collection for underrepresented classes
> * Data augmentation and synthetic oversampling strategies
> * Focal loss and class-balanced sampling methods
> * Hierarchical classification approaches for fine-grained categories
> * Cost-sensitive learning frameworks that penalize minority class errors more heavily
>
> We have updated Figures 3, 5, and 6 to improve readability. We have also removed the redundant information in the updated manuscript.

---

### Review · Reviewer_W7VZ · 2026-01-05

**Summary Of Contributions:**

This manuscript addresses a clinically meaningful problem. Cataracts are a leading cause of reversible blindness worldwide, particularly in underserved and remote regions, where access to ophthalmic care is limited. The development of a low-cost, scalable screening solution using slit lamp imaging is therefore quite relevant, and the creation of a large, publicly available dataset has the potential to significantly advance research and innovation in this area.

**Additional Comments:**

none

**Audience:**

Yes

**Audience Explanation:**

would be of interest of anyone working at the intersection of AI and ophthalmology

**Claims And Evidence:**

No

**Claims Explanation:**

The use of a portable slit lamp camera integrated with a smartphone is an interesting design choice, particularly given the goal of expanding access to care in resource-limited settings. However, image quality from smartphone-based slit lamp systems may be inferior to that obtained with traditional tabletop slit lamps. While portability offers clear advantages for deployment in remote areas, the potential trade-off in image quality should be more explicitly discussed, including how this may affect cataract detection, grading performance, and generalizability of the proposed models.

There appears to be some ambiguity regarding expert annotation. The abstract states that images were annotated by “ophthalmology experts,” yet the methods section describe only one ophthalmologist (a retina specialist), with the remaining annotators being optometrists. This distinction is important and should be clarified for transparency. In addition, it would be helpful to explain the rationale for involving a retina specialist for expert validation rather than an anterior segment surgeon, given that cataract diagnosis and grading are primarily within the anterior segment domain.

Several points in the ground-truth labeling scheme would benefit from clarification. Under cataract type, it’s unclear whether aphakia is included within the “Others” category. Additionally, under cataract grading, the label “Mild” may be misleading, as grading systems often combine early disease into a “Mild/Moderate” category; the authors may wish to clarify or adjust this terminology to better align with established clinical conventions. In the anatomical and pathological annotation section, the term “conjunctival hemorrhages” appears to be used; it would be helpful to confirm whether this refers specifically to subconjunctival hemorrhages.

**Requested Changes:**

Figure 3 depicts a slit lamp photograph obtained through a miotic and likely undilated pupil.  From a clinical standpoint, pupil dilation is generally considered the gold standard for comprehensive cataract assessment, as it allows visualization of the entire lens. The manuscript should clarify whether dilating drops were administered prior to image acquisition and, if not, discuss how undilated imaging may limit cataract evaluation and grading accuracy.

The manuscript notes that opticians were involved in ocular imaging, which raises questions regarding scope of practice. In many regions, opticians primarily function as spectacle or optical vendors and are not typically trained or credentialed to perform ocular imaging. The authors should clarify whether opticians in their region are formally trained to acquire slit-lamp photographs and how imaging consistency and quality were ensured across operators.

Figure 6 illustrates examples of poor-quality images in which the slit beam is not consistently positioned on the lens, and the lens is not always centered within the image. This raises concerns about image acquisition variability, potentially due to patient movement or operator technique. It would help if the authors include the numbers or percentages of photos that are good vs. acceptable vs. poor quality.

From a medical and clinical research perspective, abstracts typically include quantitative performance metrics to allow readers to quickly assess the effectiveness of the proposed method. In this abstract, no specific numerical results (such as AUC, etc.) are reported for the proposed AI frameworks. As a result, it is difficult to gauge the  actual performance or clinical utility of the approach based on the abstract alone. The authors may wish to clarify whether the omission of performance metrics is intentional and standard for this type of dataset-focused contribution, or whether representative results could be included. Providing key quantitative outcomes would help readers better understand how effective the proposed methods are and how they compare to existing approaches.

---

> ### Author Response · Authors · 2026-01-20
> **We thank the reviewer for providing their feedback and suggestions. Our response to the reviewer’s questions are provided below**
>
> **Impact of portable slit-lamp images on generalizability**: We acknowledge that smartphone-integrated portable slit-lamp systems may exhibit greater variability in illumination, focus, and image quality, which can influence the subsequent diagnostic performance. However, the primary objective of the CatScreen dataset is to enable scalable cataract screening and triage in resource-limited settings where traditional tabletop slit-lamps are often unavailable. To explicitly address this trade-off, we have added a discussion in the introduction section to highlight the implications of portable imaging on cataract detection and model generalizability. Furthermore, Catscreen includes explicit image-quality and gradability annotations, enabling models to identify and adapt to suboptimal inputs. The inclusion of image quality assessment in both the proposed frameworks plays a critical role in mitigating error propagation by enabling early filtering or contextual interpretation of downstream predictions. By training on a mixture of variable-quality images, the proposed benchmarks promote robustness and improve generalization beyond controlled tertiary-care imaging settings. We clarify that CatScreen is not intended to replace high-end ophthalmic diagnostic systems, but rather to support early-stage screening and referral workflows. This reinforces the relevance of the proposed models for real-world clinical and community screening applications.
>
> **Expert annotator**: In our clinical setting, image acquisition and initial annotation were performed by formally trained optometrists. These personnel undergo structured institutional training in slit-lamp operation, ocular imaging, and quality assurance prior to participation. Imaging consistency was maintained through standardized acquisition protocols, on-site supervision, and real-time quality checks during image capture before annotation. We further clarify that the validating ophthalmologist is a retina-trained clinician with over a decade of experience in slit-lamp-based anterior segment evaluation. While cataract diagnosis and surgery primarily fall within the anterior segment domain, slit-lamp-based anterior segment assessment is routinely performed across ophthalmic subspecialties. We have updated the abstract and the data annotation to include these clarifications in the revised manuscript.
>
> **Clarification on groundtruth labelling**: The primary objective of CatScreen is to identify the presence of cataract and its severity grade. Consequently, all non-cataract lens conditions, including aphakia, pseudophakia, and other non-cataract cases, are grouped under the “Others” category. Regarding the cataract grading, we acknowledge that clinical grading systems often distinguish early disease using multiple fine-grained categories. In CatScreen, the label “Mild” intentionally aggregates grades 1, 2, and 3, while the “Severe” corresponds to grades 4 and 5. This was adopted to mitigate inter-observer variability and the inherent subjective bias in fine-grained clinical grading. This is more prevalent in real-world screening environments.
> The term “conjunctival hemorrhages” refers to “subconjunctival hemorrhages”. We have updated the ground-truth collection section and the term for clinical precision in the revised manuscript.

---

> ### Author Response · Authors · 2026-01-20
>
> **Image acquisition**: Figure 3 is a sample image for showcasing the annotation process. In CatScreen, images are captured under both dilated and undilated pupil conditions, as mentioned in the image collection section. Pupil-dilating drops were administered when clinically appropriate; however, in outreach and screening scenarios, routine dilation was not feasible due to operational constraints and patient comfort. As a result, the dataset contains both dilated and undilated images. As the reviewer correctly pointed out, undilated images may limit cataract visualization; including such images is intentional to reflect real-world screening workflows. This variability enables evaluation of model robustness under practical deployment conditions.
>
> The variations in slit beam positioning and lens centering can arise due to patient movement, operator technique, and practical constraints inherent to portable slit-lamp imaging. We have reported the distribution of image quality labels in the CatScreen dataset in Table 2. Each image is categorized into good, acceptable, and poor quality based on predefined clinical criteria during annotation. CatScreen aims to reflect real-world community screening conditions, where ideal imaging remains a challenging task.
>
> **Performance metrics in abstract**: The reviewer has made an important observation regarding the inclusion of quantitative performance metrics in the abstract. These metrics represent the accuracy and effectiveness of the proposed model. In the present work, the primary contribution is the introduction of CatScreen, a large-scale, multimodal, and clinically annotated benchmark dataset for cataract screening. The AI frameworks presented in the paper serve as baseline benchmarks to demonstrate the dataset's utility and provide reference points for future algorithmic development. As discussed, there is a class imbalance across several tasks that affects performance, highlighting the need for more robust, imbalance-aware modeling strategies. Therefore, reporting the current results in the abstract could be misleading, as these results establish an initial baseline and expose open challenges rather than representing optimized performance. This approach is standard for dataset-focused contributions, where the primary value lies in the resource itself.

---

### Author Response · Authors · 2026-01-20
**Summary of Major Revisions**

1. **New Multimodal Experiments**: Tables 4, 5, and 8 now include metadata integration showing consistent performance improvements across all frameworks.
2. **Interpretability Analysis**: Added GradCAM (Figure 6a) and SHAP (Figure 6b) visualizations demonstrating clinically-aligned feature attribution.
3. **SSA Pipeline Implementation**: Table 5 presents full sequential analysis with quality gate evaluation (Table 6).
4. **Clarified Annotation Process**: Detailed three-tier annotation hierarchy and noisy label generation mechanism in Section 3.2.
5. **Updated Figures**: Improved resolution for Figures 3, 5, and 6.
6. **Future Research Directions**: Added discussion in Section 6.4 outlining research opportunities enabled by noisy-labeled, unlabeled, and segmentation subsets.
7. **Broader Impact Considerations**: Added a broader impact discussion in Section 8 covering the implications of class imbalance, demographic fairness, and human oversight in clinical deployment.

---

### Decision · Action_Editor_V64d · 2026-03-02

**Recommendation:** Accept with minor revision

**Additional Comments:**

Although this paper is a solid dataset paper, there are remaining concerns that the interpretability claims are overstated, incomplete use/exploration of the dataset’s features, and limited evaluation under class imbalance. Consequently, all reviewers are "leaning accept" and the paper is recommended for conditional acceptance based on the dataset contribution, but the final version should include clearer positioning regarding these aspects, and more restrained claims, especially where it relates to clinical aspects and impact.

In particular, the reviewers made the following comments during discussion of this paper that are directly relevant to its revision. In particular, note that each reviewer said that the revision only partially addressed his or her concerns:

"The authors addressed some of my concerns in the rebuttal, although I am still not particularly convinced by the interpretability impact offered by the sequential framework itself, considering the results shown are more-so evaluating the trained models themselves through existing techniques like GradCAM. To this end, I think the writing could be made more clear and these sections in the results and discussions expanded. Regardless, I think the dataset itself could be of use to researchers in this field."

"The clarifications provided regarding the generation of the noisy label set, the integration of metadata in the multimodal experiments, and the additional interpretability analyses partially address my initial concerns. I appreciate the effort to make these aspects more transparent and empirically supported in the revised manuscript."

"Authors were very responsive to my comments and made edits that clarified the clinical practices used in data collection and rationale behind decisions made in the data collection process. While I still have concerns about the effectiveness of this screening method [in clinical settings], I am satisfied that the paper contains other contributions that are of interest."

**Audience:**

Yes

**Audience Explanation:**

Yes, the dataset is a nice contribution that will likely benefit the community.

**Claims And Evidence:**

Yes

**Claims Explanation:**

The main contribution of this paper, the dataset, is well-supported. The reviewers largely agree that its scale, quality, and analysis are clearly demonstrated after the revision. However, there remain some concerns about the interpretability, clinical effectiveness, and robustness to real-world conditions as described in the paper, which are only partially substantiated. Similarly, some aspects of the dataset are not fully explored in the experiments. Consequently, while the dataset is likely to be a good contribution to the community, some of the methodological, clinical, and interpretability claims could benefit from more empirical support or more tempered framing.